# Soluble Fas ligand drives autoantibody-induced arthritis by binding to DR5/TRAIL-R2

**Dongjin Jeong[1,2†], Hye Sung Kim[2†], Hye Young Kim[3], Min Jueng Kang[4,5], Hyeryeon Jung[4,5], Yumi Oh[6], Donghyun Kim[2], Jaemoon Koh[1], Sung-Yup Cho[6], Yoon Kyung Jeon[1], Eun Bong Lee[7], Seung Hyo Lee[8], Eui-Cheol Shin[8], Ho Min Kim[8], Eugene C Yi[4,5], Doo Hyun Chung[1,2]\***

[1]Department of Pathology, Seoul National University College of Medicine, Seoul, Republic of Korea; [2]Laboratory of Immune Regulation in Department of Biomedical Sciences, Seoul National University College of Medicine, Seoul, Republic of Korea; [3]Department of Biomedical Sciences, Seoul National University College of Medicine, Seoul, Republic of Korea; [4]Department of Molecular Medicine and Biopharmaceutical Sciences, School of Convergence Science, Seoul, Republic of Korea; [5]Technology and College of Medicine or College of Pharmacy, Seoul National University, Seoul, Republic of Korea ; [6]Department of Biochemistry and Molecular Biology, Seoul National University College of Medicine, Seoul, Republic of Korea; [7]Department of Internal Medicine, Seoul National University College of Medicine, Seoul, Republic of Korea; [8]Graduate School of Medical Science and Engineering, Korean Advanced Institute of Science and Technology (KAIST), Daejeon, Republic of Korea

**\*For correspondence:**
doohyun@snu.ac.kr

[†]These authors contributed equally to this work

**Competing interests:** The authors declare that no competing interests exist.

**Abstract** To date, no study has demonstrated that soluble Fas ligand (sFasL)-mediated inflammation is regulated via interaction with Fas in vivo. We found that FasL interacts specifically with tumor necrosis factor receptor superfamily (TNFRSF)10B, also known as death receptor (DR)5. Autoantibody-induced arthritis (AIA) was attenuated in FasL (*Fasl^gld/gld*)- and soluble FasL (*Fasl^Δs/Δs*)-deficient mice, but not in Fas (*Fas^lpr/lpr* and *Fas^−/−*)- or membrane FasL (*Fasl^Δm/Δm*)-deficient mice, suggesting sFasL promotes inflammation by binding to a Fas-independent receptor. Affinity purification mass spectrometry analysis using human (h) fibroblast-like synovial cells (FLSCs) identified DR5 as one of several proteins that could be the elusive Fas-independent FasL receptor. Subsequent cellular and biochemical analyses revealed that DR5 interacted specifically with recombinant FasL–Fc protein, although the strength of this interaction was approximately 60-fold lower than the affinity between TRAIL and DR5. A microarray assay using joint tissues from mice with arthritis implied that the chemokine CX3CL1 may play an important downstream role of the interaction. The interaction enhanced *Cx3cl1* transcription and increased sCX3CL1 production in FLSCs, possibly in an NF-κB-dependent manner. Moreover, the sFasL–DR5 interaction-mediated CX3CL1–CX3CR1 axis initiated and amplified inflammation by enhancing inflammatory cell influx and aggravating inflammation via secondary chemokine production. Blockade of FasL or CX3CR1 attenuated AIA. Therefore, the sFasL–DR5 interaction promotes inflammation and is a potential therapeutic target.

## Introduction

The Fas/Fas ligand (FasL) pathway is a major regulator of cell-mediated apoptosis and immune-privileged status in the eyes and testes (*Bellgrau et al., 1995*; *Griffith et al., 1995*) and plays a pivotal role in maintaining immune tolerance to autoantigens. Membrane-bound FasL (mFasL) can be proteolytically cleaved by matrix metalloproteinases (MMPs), generating soluble FasL (sFasL) (*Kayagaki et al., 1995*). mFasL is essential for triggering Fas-induced apoptosis (*O' Reilly et al., 2009*). By contrast, sFasL regulates non-apoptotic processes (*Seino et al., 1998*) but stimulates apoptosis in fibroblast-like synovial cells (FLSCs) of patients with rheumatoid arthritis in a dose-dependent manner (*Kim et al., 2007*). Therefore, mFasL and sFasL have different functions in vivo that are not completely understood (*O' Reilly et al., 2009*).

FasL-mediated apoptotic and non-apoptotic cellular pathways regulate biological processes via interaction with Fas. Several studies have demonstrated that sFasL acts as a Fas-dependent chemotactic agent for neutrophils in non-apoptotic pathways (*Ottonello et al., 1999*; *Seino et al., 1998*), and sFasL levels are increased in patients with autoimmune diseases, graft-versus host disease, and cancer (*Das et al., 1999*; *Hashimoto et al., 1998*; *Murayama et al., 1999*). Therefore, sFasL may be involved in stimulating inflammation. However, whether sFasL-mediated inflammation is regulated via interaction with Fas in vivo and how this regulates inflammation in various microenvironments remains unclear. We identified a Fas-independent membrane-bound receptor for FasL and investigated its function. In this study, we demonstrate that death receptor (DR)five is a Fas-independent membrane-bound receptor for FasL and that the sFasL-DR5 interaction increases inflammation via the CX3XL1–CX3CR1 axis.

## Results

### DR5 is a Fas-independent receptor for sFasL that promotes arthritis

To investigate the Fas-independent function of FasL in the arthritis model, we injected wild-type (WT), *Fas^{lpr/lpr}*, *Fasl^{gld/gld}*, and *Fas^{−/−}* mice with K/BxN serum (*Ji et al., 2001*). The WT, *Fas^{lpr/lpr}, and Fas^{−/−}* mice developed autoantibody-induced arthritis (AIA), whereas the *Fasl^{gld/gld}* mice exhibited attenuation of joint swelling and expression of *Il6*, *Tnfa*, *Il1b*, *Ccl2*, and *Ccl3* (*Figure 1A*, *Figure 1—figure supplement 1A–C*). In addition, anti-FasL antibody, but not anti-Fas antibodies, attenuated AIA and joint pro-inflammatory cytokine expression in WT mice (*Figure 1—figure supplement 1D, E*). K/BxN serum transfer increased sFasL in the synovial fluid of WT and *Fasl^{Δm/Δm}*, but not *Fasl^{Δs/Δs}* mice (*Figure 1—figure supplement 1F*). *Fasl^{Δs/Δs}* and *Fasl^{gld/gld}*, but not *Fasl^{Δm/Δm}*, mice exhibited minimal joint swelling and cytokine expression. Administration of recombinant (r) sFasL induced joint inflammation in *Fasl^{Δs/Δs}* and *Fasl^{gld/gld}* mice (*Figure 1B–D*, *Figure 1—figure supplement 1G–I*). Furthermore, rsFasL injection aggravated joint inflammation in both *Fas^{−/−}* mice and WT mice (*Figure 1—figure supplement 1J*). Adoptive transfer of splenocytes from WT, but not *Fasl^{gld/gld}*, mice also induced arthritis in *Fasl^{gld/gld}* mice (*Figure 1—figure supplement 1K*). Taken together, these findings suggest that sFasL generation in hematopoietic cells might be a possible biological candidate to promote AIA by binding to a Fas-independent receptor. Consequently, we performed affinity purification–mass spectrometry (AP-MS) analyses using human (h) FLSCs (*Figure 1—figure supplement 2A,B*; *Table 1*). The results imply that DR5, which is encoded by the tumor necrosis factor receptor superfamily (TNFRSF)10B gene, may be a Fas-independent FasL receptor (*Figure 1E*). DR5 was expressed in synovial non-immune cells in mice with AIA, rather than in leukocytes, in mice with AIA. DR5 was also expressed in hFLSCs and was more abundant in joint tissues from patients with rheumatoid arthritis than in tissues from healthy control subjects (*Figure 1F,G*, *Figure 1—figure supplement 2C and D*). In addition, biotinylated recombinant hFasL bound to EL4 mouse T cells expressing hDR5. This interaction was blocked by anti–hDR5 antibodies or recombinant human (h) TNF-related apoptosis-inducing ligand (TRAIL). Biotinylated hTRAIL bound to EL4 mouse T cells expressing hDR5 but not to those expressing hFas (*Figure 1H*, *Figure 1—figure supplement 2F–I*). Furthermore, an anti-Fas or anti-DR5 antibody blockade partially inhibited the binding of hFasL–Fc protein to the cell surface of hFLSCs (*Figure 1—figure supplement 2E*). Anti-DR5 and Fas antibodies did not show cross-reactivity between human and mouse DR5 and Fas, respectively (*Figure 1—figure supplement 3A–D*). Binding of hFasL–Fc protein to the cell surface was completely inhibited by knockdown of *TNFRSF10B* and *FAS*, but not other TNFRSFs such as *TNFRSF1A* (TNFR1),

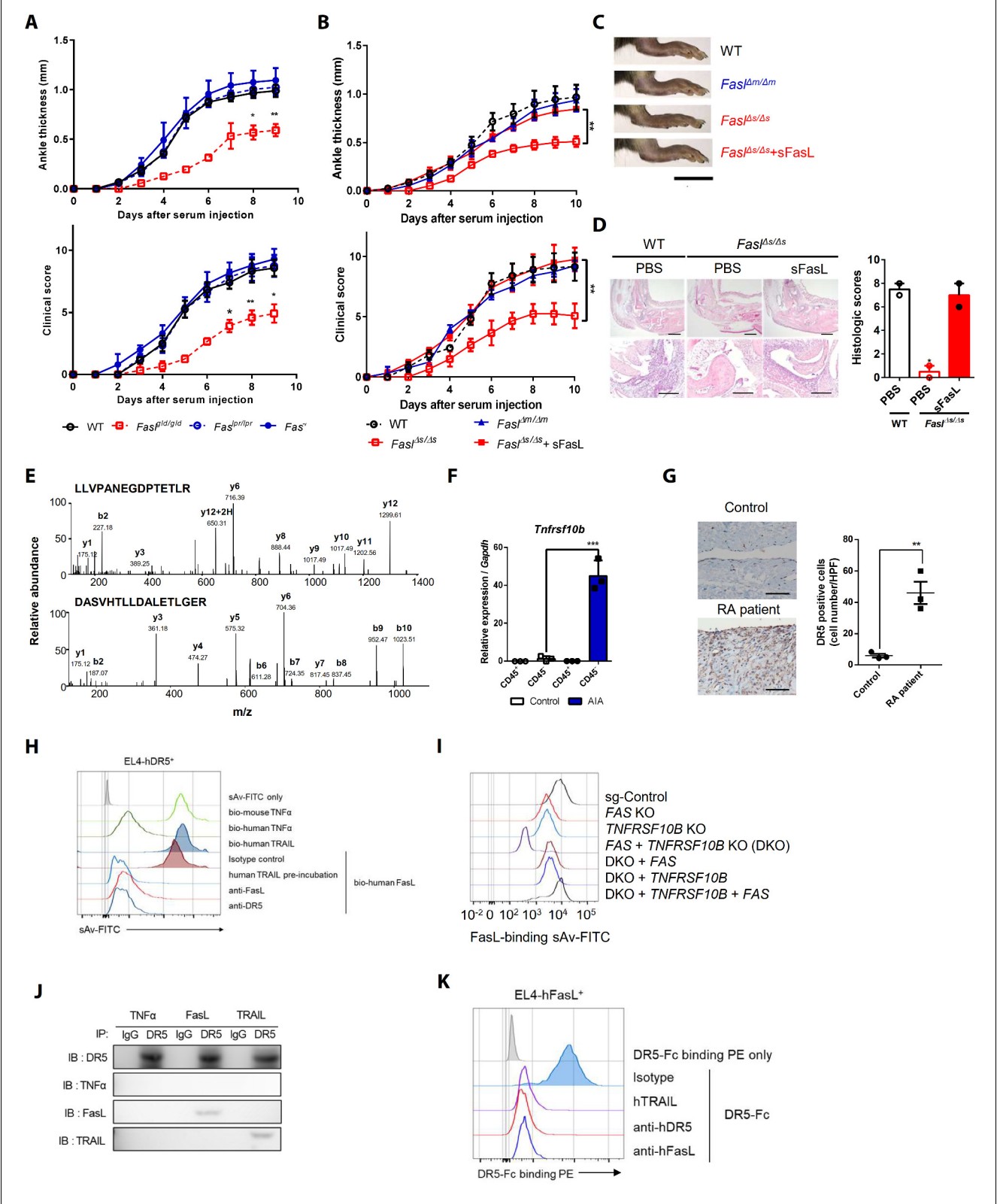

**Figure 1.** Death receptor (DR5) is a Fas-independent receptor for soluble Fas ligand (sFasL) that promotes arthritis. (**A**) Joint swelling and clinical scores in wild-type (WT), *Fas^lpr/lpr*, *Fasl^gld/gld*, and *Fas^−/−* mice (n = 6 per group). (**B**) Joint swelling and clinical scores in WT, *Fasl^Δm/Δm*, *Fasl^Δs/Δs*, and *Fasl^Δs/Δs* mice injected with sFasL (n = 6 per group). (**C, D**) Gross and microscopic examination of arthritis (magnified 10× in the upper panel and 200× in the lower panel). Scale bars: 1 cm (**C**), 200 μm (**D**, upper panel), and 100 μm (**D**, lower panel). (**E**) Tandem mass spectra of unique DR5 peptides. (**F**)

*Figure 1 continued on next page*

Figure 1 continued

Transcript levels of *Tnfrsf10b* in synovial CD45$^+$ immune cells and CD45$^-$ non-immune cells from WT mice with or without AIA. (G) Immunohistochemistry of DR5 expression in joint tissue from a healthy control subject and a patient with rheumatoid arthritis (n = 3; magnified 400×, scale bar: 50 μm). (H) Flow cytometric analysis of biotinylated protein binding to EL4 cells transfected with human WT *TNFRSF10B* preincubated with recombinant hTRAIL, or simultaneously incubated with anti-FasL, or anti-DR5 antibodies. (I) Flow cytometric analysis of biotinylated FasL binding on hFLSCs with *FAS* and/or *TNFRSF10B* knockout, and *TNFRSF10B* and/or *FAS* overexpression in *FAS* and *TNFRSF10B* double knockout (DKO) cells. (J) hLFSCs were preincubated with TNF-α (as a negative control), FasL, or TRAIL and cross–linked with BS$^3$. Lysates from these cells were immunoprecipitated with anti–DR5 or control IgG antibody and immunoblotted with anti-DR5, TNF-α, FasL, or TRAIL antibodies. (K) Flow cytometric analysis of DR5–Fc binding on EL4 cells transfected with human WT *FASLG* in the presence of recombinant hTRAIL, anti-DR5, or FasL antibodies. Data were pooled from three (A, B, and D–G) or four (H, K) independent experiments and are presented as mean ± standard error of the mean (SEM). *p<0.05; **p<0.01; ***p<0.005. Data were analyzed using one-way analysis of variance (ANOVA).

The online version of this article includes the following source data and figure supplement(s) for figure 1:

**Source data 1.** Numerical data obtained during experiments represented in *Figure 1*, *Figure 1—figure supplements 1*, *3,* and *4*.
**Figure supplement 1.** Gross examination and measurement of soluble Fas ligand (sFasL) expression in the joints of mice with autoantibody-induced arthritis (AIA).
**Figure supplement 2.** Schematic diagram showing affinity purification–mass spectrometry (AP–MS) analyses, death receptor (DR) 5 and Fas expression in mouse and human joint cells, and assessment of FasL binding to DR5 using flow cytometry.
**Figure supplement 3.** Evaluation of cross-reactivity and functional activity of purchased reagents.
**Figure supplement 4.** Comparison of sFasL and DR5 binding with that of TRAIL and other TNFRSFs based on flow cytometry, co-immunoprecipitation, and surface plasmon resonance assays.

---

*TNFRSF10A* (DR4), and *TNFRSF12* (DR3) in the presence of anti-Fas antibodies (*Figure 1—figure supplement 4A,B*). Furthermore, knockout (KO) of the *FAS* or *TNFRSF10B* gene in hFLSCs by CRISPR/Cas9 gene editing (*Figure 1—figure supplement 4C,D*) decreased binding of biotinylated sFasL to cell surfaces (*Figure 1I*), and binding was completely abolished in *FAS* and *TNFRSF10B* double knockout (DKO) hFLSCs. Re-expression of DR5 gene in DKO hFLSCs rescued biotinylated sFasL and sTRAIL binding to these cells, whereas biotinylated sFasL, but not biotinylated sTRAIL, bound to DKO cells re-expressing the *FAS* gene (*Figure 1I*, *Figure 1—figure supplement 4D,E*). Furthermore, specific binding of DR5 to FasL was confirmed by surface plasmon resonance ($K_D$: $1.23 \times 10^{-12}$ M for DR5–FasL versus $6.01 \times 10^{-13}$ M for DR5–TRAIL; *Figure 1—figure supplement 4F,G*) and immunoprecipitation of lysates from sFasL-treated hFLSCs using anti-DR5 antibodies or anti-His antibodies to His-tagged sFasL (*Figure 1J*, *Figure 1—figure supplement 4H*). Meanwhile, rhDR5–Fc protein binding to EL4 cells expressing WT hFasL was inhibited by treatment with recombinant hTRAIL, anti-hDR5, or anti-hFasL antibodies, indicating that DR5 can interact with both mFasL and sFasL (*Figure 1K*, *Figure 1—figure supplement 4I–L*). Collectively, these findings indicate that DR5 is a Fas-independent receptor for both mFasL and sFasL.

## FasL and TRAIL compete for DR5 binding and exert similar effects on cell death

TRAIL (TNFSF10) is a specific ligand for DR5 (*Walczak et al., 1997*). Preincubation with rhFasL, but not rhTRAIL, decreased rhFasL–Fc protein binding to EL4 cells expressing hFas (*Figure 2A*, *Figure 1—figure supplement 2G*), whereas preincubation with rhTRAIL or rhFasL inhibited rhFasL–Fc protein binding to hFLSCs and hDR5-expressing EL4 cells (*Figure 2B*, *Figure 1—figure supplement 2F and J*). Incubation with excess sTRAIL or sFasL inhibited the sFasL–DR5 (lanes 2 and 3) and sTRAIL–DR5 (lanes 5 and 6) interactions in hFLSCs treated with low concentrations of sFasL or sTRAIL, respectively. (*Figure 2—figure supplement 1A*). These findings indicate that TRAIL and FasL compete for binding to DR5. The crystal structures of the FasL/DcR3 (Protein Data Bank: 4 MSV) and TRAIL/DR5 (1D4V and 1DU3) complexes show that FasL forms a trimer similar to other tumor necrosis factor ligands, and DcR3 or DR5 binds to the interface formed by two adjacent FasL or TRAIL monomers stoichiometrically in a ratio of 3:3 (FasL:DcR3 or TRAIL:DR5; *Figure 2—figure supplement 1B*; *Cha et al., 2000*; *Liu et al., 2016*; *Mongkolsapaya et al., 1999*). Moreover, superimposition of these two complexes demonstrates that the interactions are similar. To test whether the binding mechanisms of FasL and TRAIL to DR5 are similar, we mutated amino acids in cysteine-rich domains (CRDs) 2 and 3 of DR5, which are critical for the TRAIL–DR5 interaction (*Figure 2C*, *Figure 2—figure supplement 1C*). In contrast to WT hDR5, the rhFasL–Fc protein did not bind to EL4 cells expressing hDR5 with mutations in CRD2 or CRD3 (*Figure 2D*), although the expression levels

**Table 1.** The list of proteins obtained from AP-MS experiment.

| Gene | Description | Location | Family |
|---|---|---|---|
| JUP | Junction plakoglobin | *Plasma Membrane* | Other |
| HBA1/HBA2 | Hemoglobin, alpha 1 | *Extracellular Space* | Transporter |
| FASLG | Fas ligand (TNF superfamily, member 6) | *Extracellular Space* | Cytokine |
| AFP | Alpha-fetoprotein | *Extracellular Space* | Transporter |
| PKP1 | Plakophilin 1 | *Plasma Membrane* | Other |
| TNFRSF10B | Tumor necrosis factor receptor superfamily, member 10b | *Plasma Membrane* | Transmembrane receptor |
| LAMA3 | Laminin, alpha 3 | *Extracellular Space* | Other |
| EPHA4 | EPH receptor A4 | *Plasma Membrane* | Kinase |
| LTF | Lactotransferrin | *Extracellular Space* | Peptidase |
| ABCB1 | ATP-binding cassette, sub-family B (MDR/TAP), member 1 | *Plasma Membrane* | Transporter |
| LRP2 | Low-density lipoprotein receptor-related protein 2 | *Plasma Membrane* | Transporter |
| RIMS1 | Regulating synaptic membrane exocytosis 1 | *Plasma Membrane* | Other |
| PTPRD | Protein tyrosine phosphatase, receptor type, D | *Plasma Membrane* | Phosphatase |
| ATRN | Attractin | *Extracellular Space* | Other |
| ADAM30 | ADAM metallopeptidase domain 30 | *Plasma Membrane* | Peptidase |
| HLA-B* | Major histocompatibility complex, class I, B | *Plasma Membrane* | Transmembrane receptor |
| KCNC2 | Potassium voltage-gated channel, Shaw-related subfamily, member 2 | *Plasma Membrane* | Ion channel |
| SPPL2A | Signal peptide peptidase like 2A | *Plasma Membrane* | Peptidase |
| BSN | Bassoon presynaptic cytomatrix protein | *Plasma Membrane* | Other |
| PTPRG | Protein tyrosine phosphatase, receptor type, G | *Plasma Membrane* | Phosphatase |
| LRRC23 | Leucine-rich repeat containing 23 | *Plasma Membrane* | Other |
| LTBP3 | Latent transforming growth factor beta binding protein 3 | *Extracellular Space* | Other |
| PLA2R1 | Phospholipase A2 receptor 1, 180 kDa | *Plasma Membrane* | Transmembrane receptor |
| OGFR | Opioid growth factor receptor | *Plasma Membrane* | Other |
| MET | MET proto-oncogene, receptor tyrosine kinase | *Plasma Membrane* | Kinase |
| NLGN2 | Neuroligin 2 | *Plasma Membrane* | Enzyme |
| CD70 | CD70 molecule | *Extracellular Space* | Cytokine |
| HLA-A* | Major histocompatibility complex, class I, A | *Plasma Membrane* | Other |
| SPTBN1* | Spectrin, beta, non-erythrocytic 1 | *Plasma Membrane* | Other |

of WT and mutant hDR5 were similar (*Figure 2—figure supplement 1D*). Moreover, rhDR5–Fc protein binding was abolished in EL4 cells expressing mutated FasL, which inhibited the interaction between FasL and DcR3 (*Figure 2E*, *Figure 2—figure supplement 1E*). Collectively, these findings indicate that the regions of DR5 that bind to hFasL overlap with those that bind to hTRAIL and that FasL binds to DcR3 and DR5 by similar mechanisms, although the precise sites of interaction between hFasL and hDR5 remain unclear. Moreover, the sFasL–DR5 and sTRAIL–DR5 interactions induced apoptosis and necroptosis in hFLSCs, but cell death was inhibited by NSCI (caspase three inhibitor) and GSK'872 (receptor-interacting serine/threonine-protein kinase three inhibitor), respectively. FasL-induced apoptosis and necroptosis were partially inhibited in Fas or DR5 gene KO

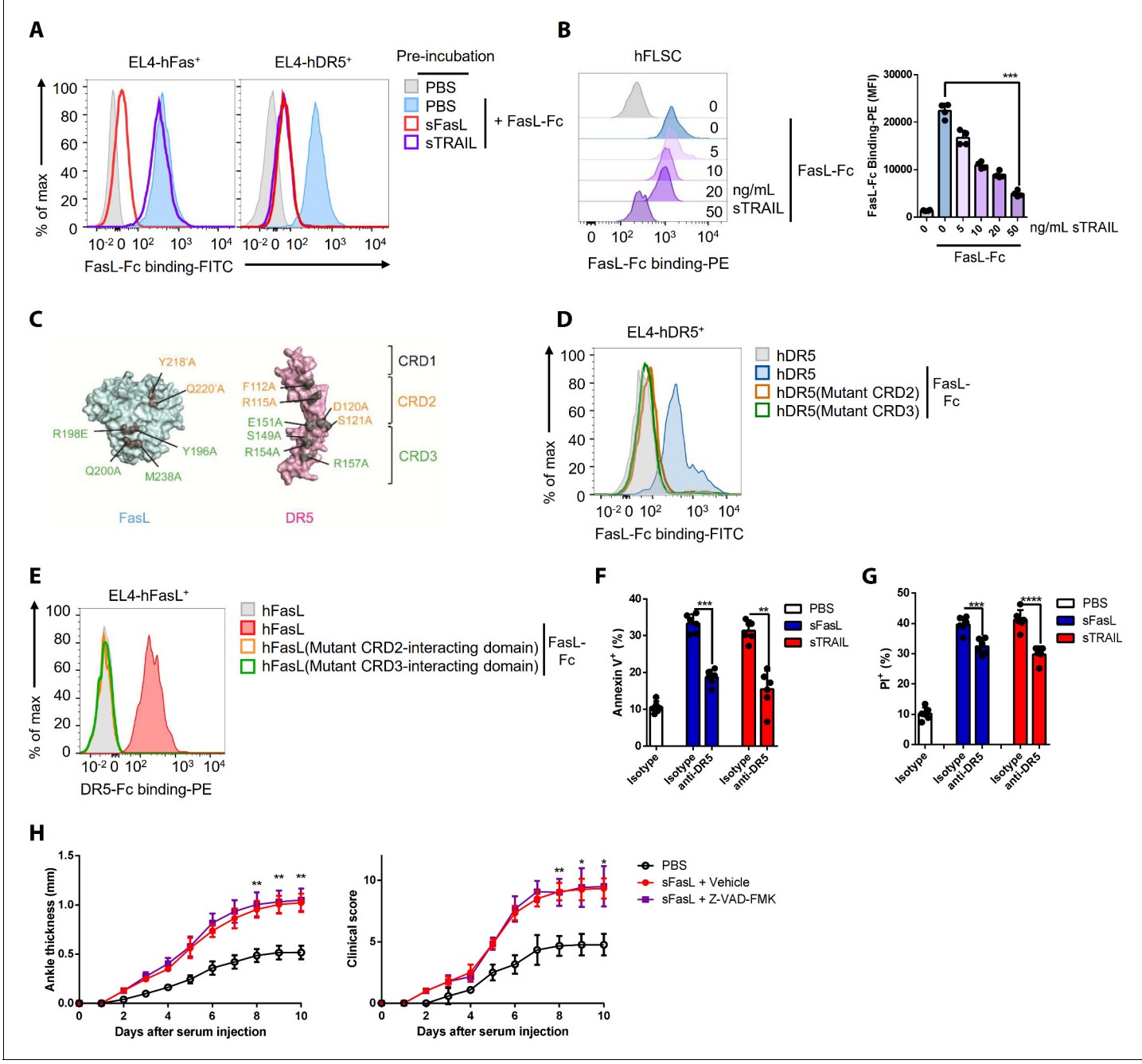

**Figure 2.** FasL and TRAIL compete for DR5 binding and exert similar effects on cell death. (**A**, **B**) FasL–Fc binding to hFLSCs or EL4 cells transfected with human *FAS*, or *TNFRSF10B* after preincubation with human sTRAIL or sFasL. (**C**) Model of FasL and DR5 derived from the crystal structure of the FasL/DcR3 complex (Protein Data Bank: 4 MSV) and TRAIL/DR5 complex (Protein Data Bank: 1D4V). (**D**, **E**) Flow cytometric analysis of FasL–Fc or DR5–Fc binding to EL4 cells transfected with human WT or mutated *TNFRSF10B* or *FASLG*. (**F**, **G**) Comparison of the effects of sFasL and sTRAIL on (**F**) apoptosis and (**G**) necroptosis in hFLSCs. (**H**) Joint swelling and clinical scores in *Fasl*^gld/gld^ mice injected with Z–VAD–FMK and/or sFasL (n = 6 per group). Data were pooled from four (**A**, **B**, and **D–G**) or three (**H**) independent experiments and are presented as mean ± SEM. *p<0.05; **p<0.01; ***p<0.005. Data were analyzed using one-way ANOVA.

The online version of this article includes the following source data and figure supplement(s) for figure 2:

**Source data 1.** Numerical data obtained during experiments represented in *Figure 2*, *Figure 2—figure supplement 3*.

**Figure supplement 1.** Structural models of the TRAIL/DR5 and FasL/DcR3 complexes, sequence alignments, and transfection efficiencies to investigate interactions involving mutant DR5 and FasL proteins.

**Figure supplement 2.** FasL-mediated apoptosis and necroptosis in hFLSCs upon treatment with specific inhibitors WT (**A**, **C**) or *FAS* and *TNFRSF10B* KO (**B**, **D**) hFLSCs were treated with sFasL in the presence of anti-Fas or anti-DR5 antibodies and cell death was quantified.

*Figure 2 continued on next page*

*Figure 2 continued*

**Figure supplement 3.** The caspase inhibitor Z–VAD–FMK does not affect AIA in WT mice but induces apoptosis in joint immune cells Joint swelling and clinical scores (A) and apoptosis (B) in joint immune cells were measured in WT mice injected with Z–VAD–FMK or the vehicle only (n = 6 per group).

hFLSCs and almost abolished in Fas plus DR5 DKO hFLSCs. These findings indicate that FasL–DR5 and FasL–Fas interactions induce apoptosis and necroptosis (*Figure 2F,G*, *Figure 2—figure supplement 2A–D*). However, administration of Z-VAD-FMK, a caspase inhibitor, did not alter AIA in WT or *Fasl*^gld/gld^ mice injected with sFasL, although cell death of immune cells in the joints was inhibited in WT mice with arthritis (*Figure 2H*, *Figure 2—figure supplement 3A–B*). This indicates that sFasL–DR5 interaction-induced apoptosis has little effect on AIA. Taken together, these results imply that FasL and TRAIL compete for DR5 binding and exert similar effects on cell death.

## sFasL–DR5 interaction enhances CX3CL1 expression in human and mouse FLSCs

To explore the non-apoptotic mechanism by which sFasL exacerbates AIA, we performed a microarray assay using joint tissues from WT, *Fas*^lpr/lpr^, and *Fasl*^gld/gld^ mice with AIA. Gene expression patterns were similar in joint tissues from WT and *Fas*^lpr/lpr^ mice, but they differed in WT and *Fasl*^gld/gld^ mice with AIA (*Figure 3A*). Among the chemokines, CX3CL1 expression in joint tissues differed the most between WT or *Fas*^lpr/lpr^ and *Fasl*^gld/gld^ mice with AIA (*Figure 3B*). Consistently, *Cx3cl1* expression was low in joint tissues from *Fasl*^gld/gld^ or *Tnfrsf10* KO mice with AIA, whereas expression was high in joint tissues from WT, *Fas*^lpr/lpr^, and *Fas*^−/−^ mice (*Figure 3C*). CX3CL1 is expressed by macrophages and FLSCs in the synovial tissue of patients with rheumatoid arthritis and plays a critical role in animal models of arthritis (*Blaschke et al., 2003*; *Nanki et al., 2002*; *Nanki et al., 2004*; *Ruth et al., 2001*). CX3CL1 also functionally acts as two distinct forms; sCX3CL1 is generated via cleavage of mCX3CL1 (*Garton et al., 2001*; *Hundhausen et al., 2003*). Both sFasL and FasL–Fc stimulation increased the levels of *CX3CL1* transcripts and sCX3CL1 protein in culture supernatants from human and mouse FLSCs, but not in culture supernatants from mouse joint leukocytes (*Figure 3D*, *Figure 3—figure supplement 1A–C*). Anti-DR5 antibody treatment or DR5-deficiency (e.g., siRNA-mediated knockdown, CRISPR/Cas9-mediated or in vivo KO of *TNFRSF10B* or *Tnfrsf10b* in human or mouse FLSCs, respectively) inhibited sFasL-mediated expression of *CX3CL1* or *Cx3cl1* transcripts and sCX3CL1 protein production in culture supernatants from human and mouse synovial fibroblasts. In contrast, knockdown or KO of *FAS*, *TNFRSF10A*, or other members of the tumor necrosis factor receptor superfamily, as well as anti-Fas antibodies did not alter *CX3CL1* expression (*Figure 3E–G*, *Figure 3—figure supplement 1D–K*). These findings imply that the sFasL–DR5 interaction increases *CX3CL1* and *Cx3cl1* transcripts and sCX3CL1 protein production in human and mouse FLSCs.

In addition, sFasL-mediated increases in *CX3CL1* transcripts and sCX3CL1 protein production were suppressed by administration of IKK (BMS345541) or proteasome (NF-κB) inhibitor (MG132) but not by administration of MEK (U0126), ERK (PD980259), or p38 kinase (SB203580) inhibitors (*Figure 3H*, *Figure 3—figure supplement 1L*). Knockdown of *RELA, CHUK*, or *IKBKB* inhibited sFasL-mediated increases in *CX3CL1* transcripts and sCX3CL1 protein production in hFLSCs (*Figure 3I* and *Figure 3—figure supplement 1M*). Furthermore, sFasL treatment increased phosphorylated (*p*)-p65 and *p*-IκBα, but decreased total IκBα in hFLSCs (*Figure 3—figure supplement 1N*). Inhibition of caspase activity did not affect the levels of *CX3CL1* transcripts and CX3CL1 protein production (*Figure 3—figure supplement 1O,P*). These findings indicate that sFasL–DR5 interaction mediates stimulation of *CX3CL1* transcription and sCX3CL1 protein production in hFLSCs, and this may be dependent on the NF-κB signaling pathway.

In contrast to sFasL, sTRAIL did not alter the expression of *CX3CL1* transcripts or sCX3CL1 protein in hFLSCs and mouse synovial cells. However, preincubation with sTRAIL inhibited sFasL-mediated *CX3CL1* transcript accumulation and sCX3CL1 protein production in a dose-dependent manner (*Figure 3J,K*, *Figure 3—figure supplement 2A and B*). Furthermore, administration of sFasL, but not sTRAIL, exacerbated AIA and the expression of *Cx3cl1*, as well as pro-inflammatory cytokines and chemokines in the joints of WT or *Fasl*^gld/gld^ mice (*Figure 3—figure supplement 2C–F*). These findings indicate that sFasL and sTRAIL have differential effects on the expression of *CX3CL1* transcripts and sCX3CL1 protein production by FLSCs.

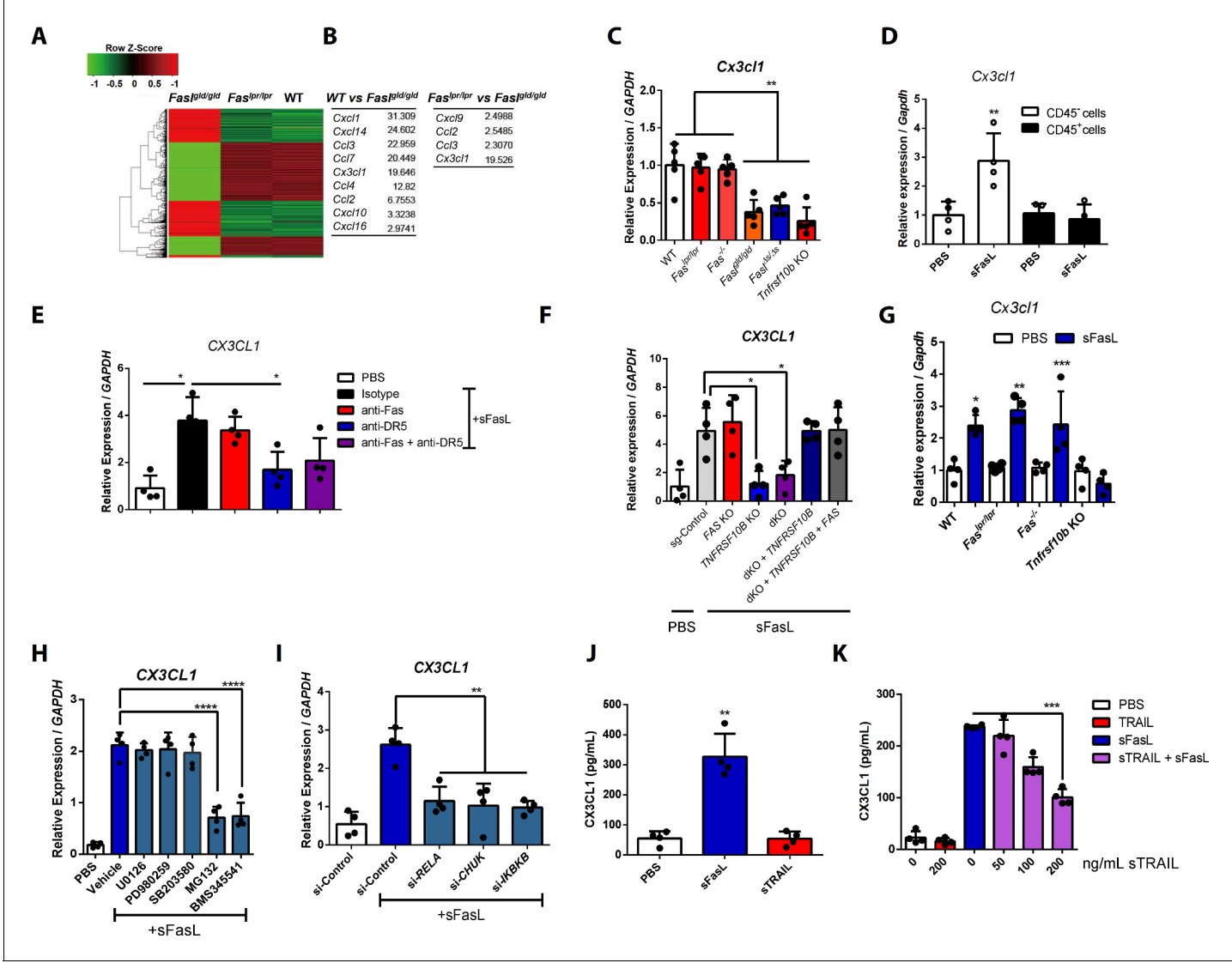

**Figure 3.** The sFasL–DR5 interaction enhances CX3CL1 expression by human and mouse FLSCs. (**A, B**) Microarray assay using joint tissues from WT, *Fas^lpr/lpr^*, and *Fasl^gld/gld^* mice with arthritis. (**C**) *Cx3cl1* transcript levels estimated in joint tissues from WT, *Fas^lpr/lpr^*, *Fas^−/−^*, *Fasl^gld/gld^*, *Fasl^Δs/Δs^*, and *Tnfrsf10b* KO mice with arthritis. (**D**) *Cx3cl1* expression in CD45^+^ immune and CD45^−^ non–immune cells from the joints of WT mice with arthritis after sFasL treatment. (**E, F**) *CX3CL1* transcript levels estimated in hFLSCs in the presence of anti-Fas and/or anti-DR5 antibodies (**E**) and *FAS* (Fas), *TNFSF10B* (DR5), or *FAS*, and *TNFRSF10B* DKO, or *TNFRSF10B* and *FAS* overexpression in DKO hFLSCs (**F**). (**G**) *Cx3cl1* expression in synovial fibroblasts from WT, *Fas^lpr/lpr^*, *Fas^−/−^*, *or Tnfrsf10b* KO mice in the presence or absence of sFasL. (**H, I**) *CX3CL1* transcript levels estimated after sFasL stimulation in hFLSCs in the presence of MEK (U0126), ERK (PD980259), p38 kinase (SB203580), and NF-κB (MG132 and BMS345541) inhibitors (**H**) or transfection with control, *RELA*, *CHUK* (IKKa), or *IKBKB* (IKKb) siRNA (**I**). (**J**) Synovial fibroblasts obtained from WT mice with arthritis were incubated with sFasL or sTRAIL and CX3CL1 levels were measured using ELISA. (**K**) hFLSCs were stimulated with sFasL after preincubation with various concentrations of sTRAIL for 30 min and CX3CL1 levels were measured in the culture supernatant. Data were pooled from three (**C–G** and **K**) or four (**H–J**) independent experiments and are presented as mean ± SEM (n = 4 for **C–K**). *p<0.05; **p<0.01; ***p<0.005. Data were analyzed using one-way ANOVA. The online version of this article includes the following source data and figure supplement(s) for figure 3:

**Source data 1.** Numerical data obtained during experiments represented in *Figure 3*, *Figure 3—figure supplements 1* and *2*.

**Figure supplement 1.** CX3CL1 expression and NF-κB signaling in FasL-treated hFLSCs and mouse synovial fibroblasts.

**Figure supplement 2.** CX3CL1 transcript expression pattern in human and mouse synovial fibroblasts, and arthritis in *Fasl^gld/gld^* and WT mice upon sFasL and sTRAIL treatment.

## The sFasL–DR5 interaction promotes joint inflammation via the CX3CL1–CX3CR1 axis

To investigate the effects of the sFasL–DR5 interaction on AIA, we administered anti-DR5 or anti-Fas antibodies to WT or *Fasl*[gld/gld] mice given sFasL. Anti-DR5, but not anti-Fas, antibodies attenuated AIA (*Figure 4A*, *Figure 4—figure supplement 1A,B*). AIA was attenuated in *Tnfrsf10b* KO mice like as *Fasl*[gld/gld] and *Fasl*[Δs/Δs] mice (*Figure 4B,C*). Moreover, administration of sFasL increased AIA in WT mice, whereas it did not in *Tnfrsf10b* KO mice. Injection of sFasL increased *Cx3cl1* transcript levels in the joints of WT, *Fasl*[Δs/Δs], and *Fasl*[Δm/Δm] mice, but not in the joints of *Tnfrsf110b* KO mice (*Figure 4D*). Injection of sCX3CL1 induced AIA in *Fasl*[gld/gld], *Fasl*[Δs/Δs], and *Tnfrsf110b* KO mice and increased the expression levels of *Ccl2*, *Ccl3*, and *Cxcl10* in joint tissues compared to control mice (*Figure 4E,F*, *Figure 4—figure supplement 1C–F*). Furthermore, AIA was attenuated more in *Cx3cr1* KO than in WT mice, but was not exacerbated by sFasL administration (*Figure 4G*). Among the inflammatory cells in the joints, macrophages and T cells expressed CX3CR1, whereas neutrophils and eosinophils did not (*Figure 5A*, *Figure 5—figure supplement 1*). CX3CL1 stimulation increased the migration and production of CCL2, CCL3, and CXCL10 in human and mouse CX3CR1[+] cells. This was inhibited by CX3CR1 blockade in mouse cells, whereas sFasL did not alter the migration of these cells (*Figure 5B–E*). These findings suggest that the sFasL–DR5 interaction may initiate and amplify inflammation via the CX3CL1–CX3CR1 axis. Thus, we injected anti-FasL or anti-CX3CR1 antibodies into mice with AIA to investigate any therapeutic effects on arthritis. We found that blockade of FasL or CX3CR1 at different phases of AIA significantly attenuated arthritis and chemokine production in the joints (*Figure 6A–F*, *Figure 6—figure supplement 1A–F*). Taken together, these findings indicate that the sFasL–DR5 interaction promotes arthritis by regulating the CX3CL1–CX3CR1 axis, which increases CCL2, CCL3, and CXCL10 production by CX3CR1[+] immune cells.

## Discussion

To date, the overwhelming majority of studies have demonstrated that FasL stimulates various biological processes by interacting with Fas because Fas is reportedly a receptor for FasL (*Guégan and Legembre, 2018*). In contrast, we demonstrated that DR5 is a Fas-independent membrane-bound receptor for FasL that promotes inflammation. DR5 (TRAIL–R2) is one of the four membrane receptors that bind TRAIL and is expressed in cancer cells and various tissues (*Wilson et al., 2009*). After binding to TRAIL, DR5 recruits an adaptor molecule via death domain interactions and induces the formation of the death-inducing signaling complex, resulting in cellular apoptosis (*Wilson et al., 2009*). The TRAIL–DR5 interaction also simulates cell activation, maturation, motility, and proliferation (*Cullen and Martin, 2015*). Similarly, our experiments showed that the sFasL–DR5 interaction induced apoptosis and non-apoptotic responses in FLSCs, implying that the biological effects of sFasL in vitro and in vivo may be attributable to interaction with either Fas or DR5. Soluble DcR3 also bind to FasL, inhibiting the function of FasL in vivo (*Pitti et al., 1998*). Furthermore, several studies have demonstrated that endogenous DcR3 attenuates the Th1 response and skews the differentiation of tumor-associated macrophages, whereas DcR3–Fc modulates dendritic cell maturation from CD14[+] monocytes (*Hsu et al., 2002*; *Hsu et al., 2005*; *Tai et al., 2012*). These findings suggest that sFasL-mediated inflammation in mice is attributable to sFasL–DcR3 interaction. However, this is impossible because DcR3 is expressed in humans, but not in mice (*Lin and Hsieh, 2011*). We suggest that FasL plays both DR5- and Fas-dependent roles in regulating various biological events in vivo, which is a paradigm shift in the biology of the classic Fas–FasL pathway.

TNFRSF proteins enhance the production of various pro-inflammatory cytokines and chemokines in an NF-κB-dependent manner (*Cullen and Martin, 2015*). Similarly, sTRAIL treatment or overexpression of DR5 or DR4 induces the secretion of inflammatory cytokines (*Tang et al., 2009a*; *Tang et al., 2009b*). However, no report has described the effect of DR5 on CX3CL1 production. CX3CL1, which is expressed in endothelial cells and FLSCs, promotes cell survival, adhesion, chemotaxis, and migration via secondary chemokines (*Nanki et al., 2017*). In our experiments, sFasL–DR5 interaction enhanced *Cx3cl1* transcription and sCX3CL1 protein production by FLSCs in mice and humans, possibly in an NF-κB dependent manner, whereas sTRAIL–DR5 interaction did not. Thus, the sFasL–DR5 interaction, but not sTRAIL–DR5 interaction, is a potent inducer of CX3CL1 expression in FLSCs. Furthermore, sFasL–DR5 interaction-mediated CX3CL1 production by FLSCs triggers the migration and chemokine production of joint leukocytes in a CX3CR1-dependent manner,

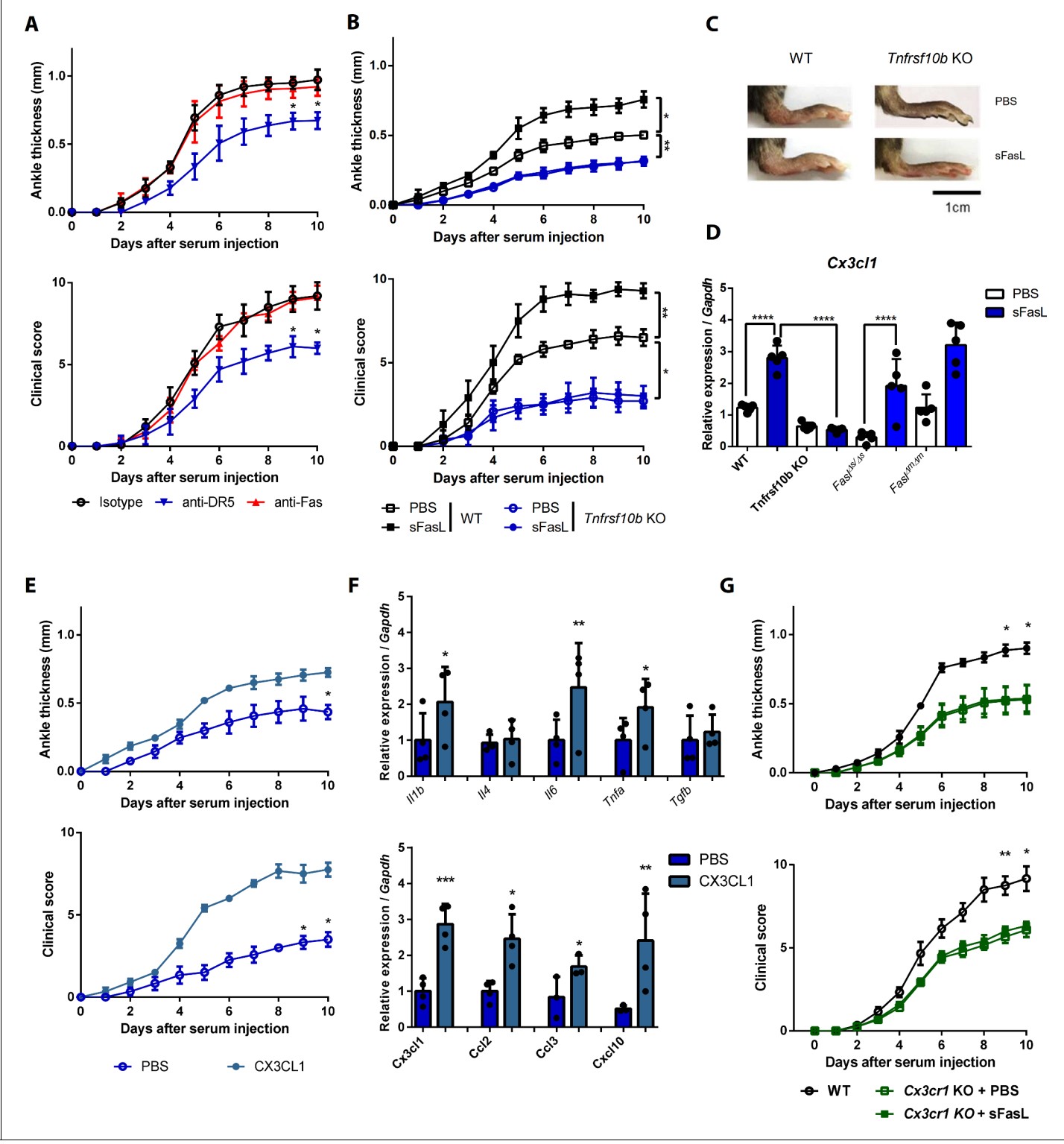

**Figure 4.** The sFasL–DR5 interaction promotes joint inflammation via the CX3CL1–CX3CR1 axis. (A) Joint swelling and clinical scores in WT mice injected with anti-DR5 or anti-Fas antibodies to measure AIA (n = 5 per group). (B, C) Joint swelling and clinical scores in WT and *Tnfrsf10b* KO mice injected with sFasL or phosphate-buffered saline (PBS) to measure AIA (n = 5 per group). (D) *Cx3cl1* transcript levels in the joints were estimated in WT, *Tnfrsf10b* KO, *Fasl*$^{\Delta s/\Delta s}$, and *Fasl*$^{\Delta m/\Delta m}$ mice injected with sFasL or PBS to measure AIA (n = 5). (E, F) Joint swelling and clinical scores (E), and transcript levels of various cytokines and chemokines in joint tissues of *Tnfrsf10b* KO mice injected with CX3CL1 or PBS to measure AIA (F) (n = 6 per group). (G) *Figure 4 continued on next page*

*Figure 4 continued*

Joint swelling and clinical scores of WT and *Cx3cr1* KO mice in the presence or absence of sFasL to measure AIA (n = 6 per group). Data were pooled from three independent experiments and are presented as mean ± SEM. *p<0.05; **p<0.01; ***p<0.005. Data were analyzed using one-way ANOVA. The online version of this article includes the following source data and figure supplement(s) for figure 4:

**Source data 1.** Numerical data obtained during experiments represented in *Figure 4*, *Figure 4—figure supplement 1*.
**Figure supplement 1.** Measurement of arthritis related to *Figure 4*.

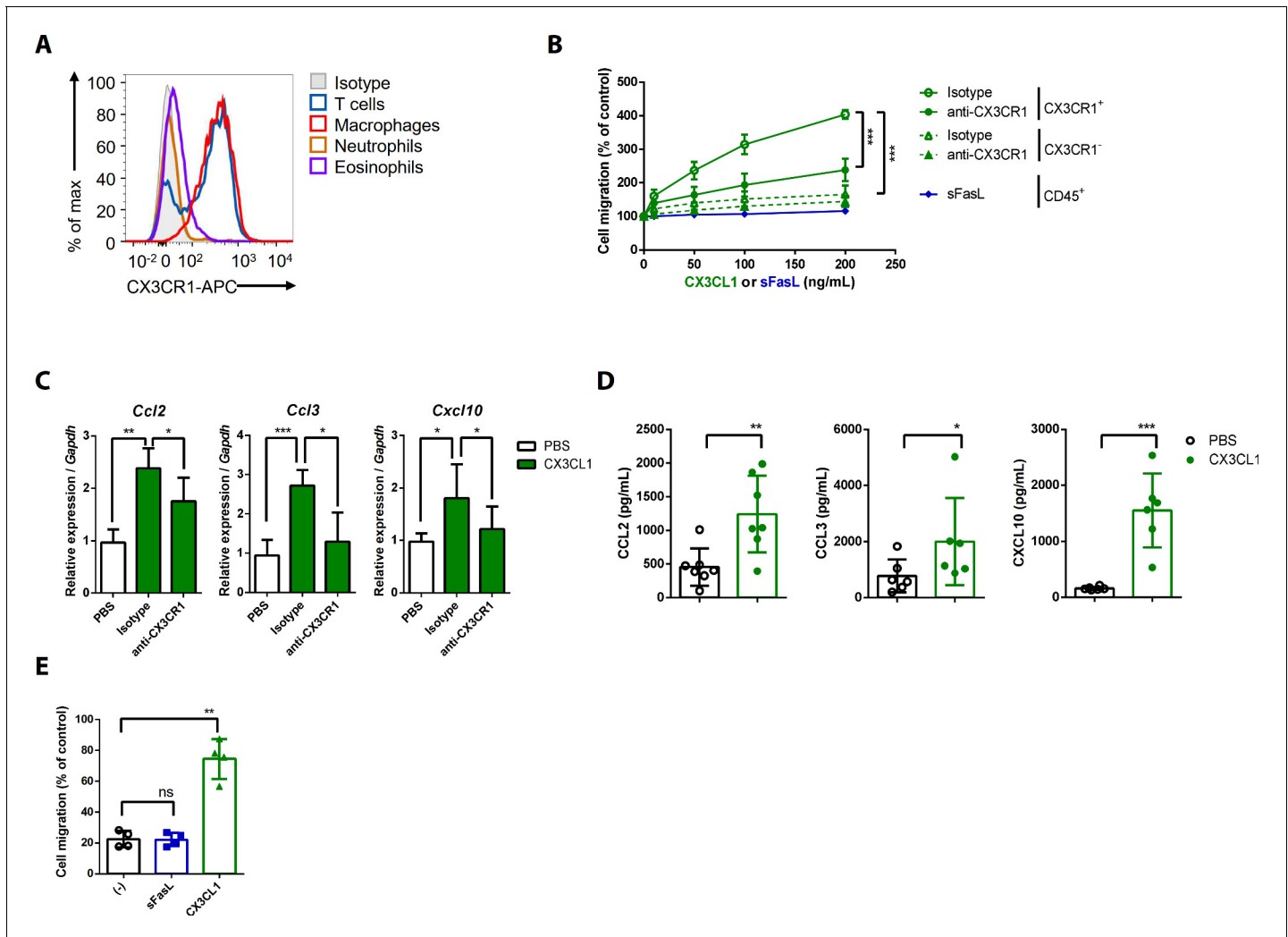

**Figure 5.** CX3CL1 stimulation increases the migration and production of chemokines in human and mouse CX3CR1⁺ cells. (**A**) Flow cytometric analysis of CX3CR1 expression in immune cells from arthritic joint tissues. (**B**) Migration assay for synovial CX3CR1⁺ or CX3CR1⁻ immune cells upon stimulation with CX3CL1 in the presence or absence of anti-CX3CR1 antibodies (green) and CD45⁺ immune cells stimulated with sFasL (blue; n = 5). (**C**) *Ccl2*, *Ccl3*, and *Cxcl10* transcript levels in synovial macrophages after stimulation with CX3CL1 (n = 6). (**D**) Levels of chemokines in supernatants of synovial leukocyte cultures from patients with rheumatoid arthritis, measured using ELISA after treatment with CX3CL1 (n = 6). (**E**) Migration assay of synovial leukocytes from patients with rheumatoid arthritis after stimulation with sFasL or CX3CL1 (n = 4). Data were pooled from three (**A**, **C**, and **D**) or four (**B**, **E**) independent experiments and are presented as mean ± SEM of independent experiments. *p<0.05; **p<0.01; ***p<0.005. Data were analyzed using one-way ANOVA.
The online version of this article includes the following source data and figure supplement(s) for figure 5:

**Source data 1.** Numerical data obtained during experiments represented in *Figure 5*.
**Figure supplement 1.** Gating strategy for flow cytometry analyses of CX3CR1 expression in immune cells from arthritic joint tissues.

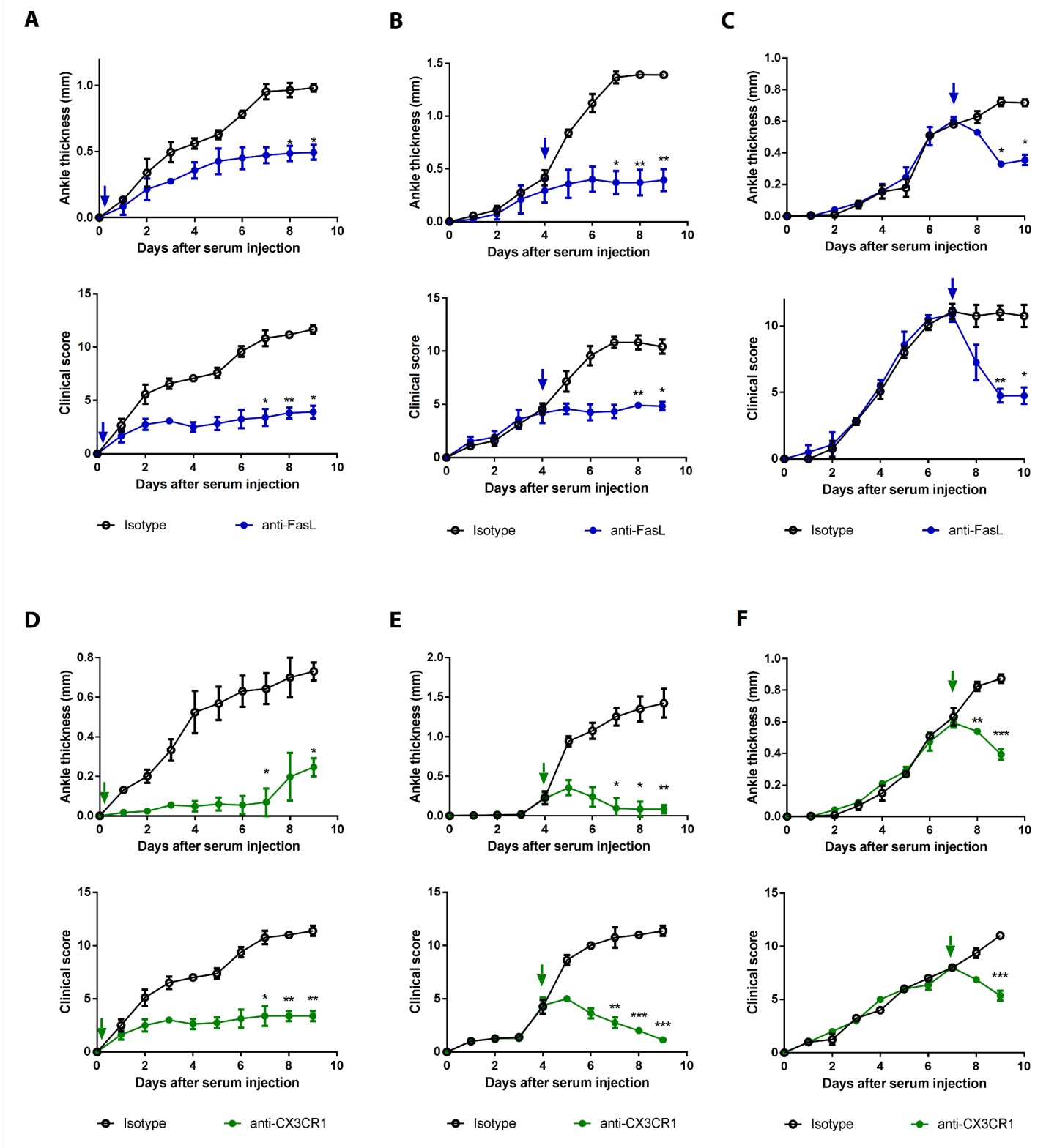

**Figure 6.** Blockade of FasL or CX3CR1 at different phases of AIA-attenuated joint inflammation. (A–F) Ankle thicknesses of WT mice injected with anti-FasL or anti-CX3CR1 antibodies (10 μg per mouse daily) on days 0–9 (A, D), days 5–9 (B, E), and days 8–10 (C, F) to measure AIA (A–C, n = 6; D–F, n = 4). The arrows in the diagrams indicate the day on which antibodies were first injected. Data were pooled from three independent experiments and are presented as mean ± SEM. *p<0.05, **p<0.01, ***p<0.005. Data were analyzed using one-way ANOVA.

*Figure 6 continued on next page*

*Figure 6 continued*

The online version of this article includes the following source data and figure supplement(s) for figure 6:

**Source data 1.** Numerical data obtained during experiments represented in *Figure 6*, *Figure 6—figure supplement 1*.

**Figure supplement 1.** Chemokine expression levels in joint tissues from WT mice injected with anti-FasL or anti-CX3CR1 antibodies at various time points.

resulting in joint inflammation. Several studies have demonstrated that sFasL acts as a potent chemotactic agent for human and mouse neutrophils at concentrations incapable of inducing cell apoptosis (*Ottonello et al., 1999*; *Seino et al., 1998*). However, our experiments demonstrated that sFasL promoted AIA via its interaction with DR5 by producing mediators, rather than acting as a chemoattractant. Thus, the sFasL–DR5 interaction may be an initiating event that amplifies inflammation via CX3CL1 production through two steps: the CX3CL1-mediated influx of inflammatory cells to target sites and the production of secondary chemokines such as CCL2, CCL3, and CXCL10 by CX3CR1$^+$ immune cells. Moreover, antibody blockade of FasL or CX3CR1 attenuated AIA in mice, suggesting that sFasL–DR5 interaction and CX3CL1–CX3CR1 axis could be therapeutic targets for controlling inflammation. In several studies, DR5 KO mice exhibited increases in tumor metastases and inflammation (*Finnberg et al., 2008*; *Zhu et al., 2014*), which is difficult to understand whether the sFasL–DR5 interaction stimulates inflammation. However, DR5 and FasL are expressed in various cell types (*O'Brien et al., 2005*; *O'Connell, 2000*; *Yuan et al., 2018*), and the sFasL–DR5 interaction likely regulates tumor surveillance and inflammation via a complex signaling network involving various biological effects that require further investigation.

The sFasL–DR5 and sTRAIL–DR5 interactions similarly induced apoptosis and necroptosis in FLSCs, and the regions of DR5 that bound FasL overlapped with those that bound TRAIL. However, sFasL binds DR5, but not DR4, whereas sTRAIL binds both DR4 and DR5, suggesting that the three-dimensional structures and binding mechanisms of interaction of sFasL and sTRAIL may differ. Therefore, although the effector functions, structures, signal transduction pathways, and downstream mechanisms involved may be similar, the two interactions may exhibit some differences. Furthermore, differences in CX3CL1 production may be attributable to differences in binding affinity, critical amino acids in DR5, or signal strength. Increasing evidence indicates that clustering of death receptors in lipid rafts plays a critical role in optimal DR5-mediated signal transduction (*Holland, 2014*). Thus, understanding how the sFasL–DR5 interaction is fine-tuned to optimize CX3CL1 expression may be necessary for identifying therapeutic targets to decrease inflammation.

In contrast to our current results, previous studies found that FasL-mediated apoptosis attenuated joint inflammation in murine arthritis models (*Guéry et al., 2000*; *Hsu et al., 2001*; *Zhang et al., 1997*). Therefore, FasL may regulate joint inflammation by different mechanisms in different arthritis models, possibly due to the distinct microenvironments of each (*Ji et al., 2002*). The K/BxN serum transfer model used in our experiments is confined to the terminal effector inflammatory responses induced by the T and B cell-independent deposition of antibodies in the joints (*Ji et al., 2002*). Thus, sFasL-mediated induction of joint inflammation may occur in the effector phase rather than the inductive phase of inflammation, although it is unclear whether this sFasL-mediated regulation is involved in the inductive phase of rheumatoid arthritis, which is dependent on T and B cells. Moreover, we did not investigate the distinct functions of sFasL versus mFasL in joint inflammation and unique functions for mFasL and sFasL may produce divergent inflammatory responses. We found that neither mFasL deficiency nor treatment with a caspase inhibitor substantially affected sFasL-mediated joint inflammation in mice. Perhaps mFasL regulates the function of leukocytes via activation-induced cell death during the inductive phase of rheumatoid arthritis, whereas an sFasL-mediated non-apoptotic mechanism promotes joint inflammation during the effector phase. In conclusion, our experiments demonstrate that DR5 is a Fas-independent membrane-bound receptor for FasL and that sFasL–DR5 interaction promotes inflammation by regulating the CX3CL1–CX3CR1 axis, which enhances production of secondary chemokines by CX3CR1$^+$ immune cells.

## Materials and methods

### Mice

The KRN TCR transgenic mouse (*Kouskoff et al., 1996*) was kindly provided by Dr. Diane Mathis and Dr. Christophe Benoist of Harvard Medical School (Boston, MA) and maintained in a C57BL/6 (B6) background (K/B). Arthritic mice (K/BxN) were obtained by crossing K/B mice with NOD (N) mice (The Jackson Laboratory, Farmington, CT). *Fas^{lpr/lpr}* and *Fasl^{gld/gld}* mice were obtained from Japan SLC (Hamamatsu, Japan). *Fas^{−/−}* (B6.129-Fas < tm1Osa>/OsaRbrc) cryopreserved embryos were provided by the RIKEN BRC (Tsukuba, Japan) via the National BioResource Project of MEXT/ AMED (Tokyo, Japan) (*Adachi et al., 1995*). The embryos were thawed and recovered at the Korea Research Institute of Bioscience and Biotechnology (KRIBB) (Ochang, Republic of Korea). *Cx3cr1*-deficient mice (B6.129-*Cx3cr1^{tm1Zm}*) were purchased from Taconic (Rensselaer, NY). *Fasl^{Δs/Δs}* and *Fasl^{Δm/Δm}* mice were a generous gift from Dr. Andreas Strasser and Dr. Lorraine O'Reilly (Walter and Eliza Hall Institute of Medical Research, Parkville, Australia). *Tnfrsf10b* KO mice were obtained from the Mutant Mouse Resource and Research Center (MMRRC, strain #030532-MU). C57BL/6 mice were purchased from Orient Bio, Inc (Seongnam, Republic of Korea). Mice were maintained in a specific pathogen-free environment at the Biomedical Research Institute of Seoul National University Hospital (BRISNUH, Seoul, Republic of Korea). Sex-matched mice aged 6–10 weeks were used for all experiments. All mouse experiments were approved by the Institutional animal care and use committee of the BRISNUH (17–0051 and 20–0171).

### Serum transfer, arthritis scoring, histological examination, and immunohistochemistry

To induce AIA, mice were injected intraperitoneally with 70 µL of sera obtained from arthritic K/BxN mice on days 0 and 2. Ankle thicknesses were measured using calipers (Manostat, New York, NY) and scored as described previously (*Kim and Chung, 2012*). Clinical scores were presented as the sum of the scores from four limbs. To examine histological alterations in the joint tissues, whole knee joints and hind paws were fixed in 10% formalin, decalcified, and embedded in paraffin. Sections were prepared from the joint tissue blocks and stained with hematoxylin and eosin. Two expert pathologists reviewed all the specimens independently. Histological analyses of joint inflammation were performed by scoring based on synovial hyperplasia, inflammatory cell infiltration, inflammatory exudate, fibroblast proliferation, subcutaneous edema, and body erosion. Each score was based on these criteria: 0, absent; 1, weak to moderate; 2, severe. Histological scores were calculated as the sum of all criteria. Formalin-fixed paraffin-embedded tissue blocks were cut into 4–µm-thick slices and analyzed immunohistochemically using anti-DR5 antibodies (ab-8416, 1:50; Abcam, Cambridge, UK) and a Benchmark XT autostainer (Ventana Medical Systems, Tuscon, AZ), in accordance with the manufacturer's instruction. DR5-positive cells in joint tissues from patients with arthritis and healthy control subjects were counted under light microscopy in three high-power fields (400× magnification). Results are presented as the mean number of DR5-positive cells per high-power field in each sample.

### In vivo experiments

Unless otherwise specified, reagents were injected into mice on days 1 and 3. A total of 1 µg of recombinant mouse sFasL, sTRAIL, or CX3CL1 (R and D Systems, Minneapolis, MN) was injected intraperitoneally and 50 µg of anti-mouse FasL, anti-Fas, anti-DR5, or anti-CX3CR1 antibodies (R and D Systems) was injected intravenously. To inhibit apoptosis in vivo, 50 µg of Z–VAD–FMK (Calbiochem, Billerica, MA,) was administered. The vehicle and isotype control injection were administered in the same way. To transfer splenocytes into *Fasl^{gld/gld}* mice, mouse spleens were homogenized and treated with red blood cell lysis solution (Qiagen, Hilden, Germany). In total, $1 \times 10^6$ spleen cells were pooled in phosphate-buffered saline (PBS) and injected intravenously into *Fasl^{gld/gld}* mice on the day before K/BxN serum injections.

### Preparation of joint cells from mice

Total cells from mouse joints were prepared at 8 days after K/BxN serum injections as described previously (*Kim and Chung, 2012*). The total joint cell preparation contained CD45+ leukocytes and

CD45⁻ non-immune synovial cells. To isolate mouse synovial cells, adherent cells were collected from joint cells cultured at 37℃ for 24 hr in RPMI 1640 media containing 10% fetal bovine serum (FBS) and 1% penicillin/streptomycin. These cells were then cultured for an additional 2 days. Western blotting analysis showed that the cells were CD45⁻ vimentin⁺ cadherin11⁺ FLSCs (data not shown). The cells were stimulated with 200 ng/mL of recombinant mouse sFasL (R and D Systems). To prepare CX3CR1⁺ cells, the total cells were labeled using allophcocyanin (APC)-conjugated anti-mouse CX3CR1 antibodies (2A9-1; Biolegend, San Diego, CA), followed by binding to anti-APC magnetic beads and enrichment using a magnetically activated cell-sorting (MACS) separator (Miltenyi Biotec, Bergisch Gladbach, Germany).

## Preparation of synovial fluid to measure soluble FasL
Synovial fluid was prepared by dissecting mouse joints in PBS and centrifuging the synovial cells at 500 g for 10 min. The synovial fluid was diluted with PBS to a concentration of approximately 1.5 mg/mL, as determined by a standard Bradford protein assay. The sFasL in the synovial fluid was measured by enzyme-linked immunosorbent assay (ELISA) using a Mouse Fas Ligand/TNFSF6 Quantikine ELISA Kit (R and D System), in accordance with the manufacturer's instructions.

Culture and stimulation of hFLSCs and isolation of mononuclear immune cells from the synovial fluid of patients with rheumatoid arthritis hFLSCs were isolated from patients with rheumatoid arthritis as described previously (*Kang et al., 2009*) and cultured in DMEM containing 10% FBS and 1% penicillin/streptomycin. Passages 5–15 of the hFLSCs were treated with 200 ng/mL recombinant human sFasL (R and D Systems) in the presence of 20 µg/mL ZB4 anti-Fas blocking antibodies (Merck, Darmstadt, Germany) to inhibit the sFasL–Fas interaction. To test the FasL–DR5 interaction, hFLSCs were treated with recombinant TRAIL (200 ng/mL or as indicated; R and D Systems) or 20 µg/mL AF631 anti-DR5 neutralizing polyclonal antibodies (R and D Systems) 1 hr prior to FasL treatment. U0126 (ERK), PD98059 (MEK), SB203580 (JNK), MG132 (NF-κB), and BMS345541 (NF-κB) were used to inhibit signaling pathways. All inhibitory reagents were purchased from Sigma-Aldrich.

To obtain synovial immune cells, synovial fluid was collected from 14 patients who had rheumatoid arthritis and fulfilled the 1987 American College of Rheumatology criteria. The fluid was washed, filtered, and then cultured in DMEM containing 10% FBS and 1% penicillin/streptomycin. Non-adherent immune cells were harvested for migration and chemokine production assays and assessed using a hemocytometer and flow cytometry. This study was approved by the Institutional Review Board of Seoul National University Hospital (Seoul, Korea; H 1009–064–332) and informed consent was obtained from all sample donors.

## Identification of a Fas-independent FasL receptor
To identify a Fas-independent FasL receptor, we implemented the in-situ chemical cross-linking method (*Kim et al., 2012*). SFasL was biotinylated with sulfo-NHS-SS biotin (Thermo Fisher Scientific, Waltham, MA), and 20 µg of biotinylated sFasL–Fc protein was incubated with $1.0 \times 10^8$ hFLSCs for 2 hr. Next, 1 mL of 3 µg/mL bis-(sulfo-succinimidyl) subsrate (BS³) cross-linking reagent was added and the mixture was kept at room temperature for 30 min. Then, we added 2 M Tris–Bis HCl pH 7.4 to a final concentration of 15 mM and allowed the cross-linking reaction to quench at room temperature for 15 min. Cells were lysed in radioimmunoprecipitation assay buffer, and biotinylated sFasL-cross-linked protein complexes were purified by avidin affinity chromatography. We used Fc protein to control for non-specific binding to the Fc region. The control samples were otherwise treated identically to the sFasL–Fc protein and Fc protein-incubated samples. The purified sFasL cross-linked protein complexes were separated by 4–12% SDS-PAGE and stained for spectrometry analyses (*Shevchenko et al., 2006* ). The peptide samples extracted from the in-gel digestions were suspended in 0.1% formic acid, loaded onto an EASY-Spray Column (15 cm × 50 µm ID, C18 stationary phase), and separated using a 5–40% gradient of 0.1% formic acid in acetonitrile for 90 min at a flow rate of 300 nL/min. Mass spectra were recorded using a Q Exactive mass spectrometer (Thermo Fisher Scientific) interfaced with an Ultimate 3000 HPLC system. The raw data were searched against the Uniprot human database (143,673 entries, June 2014 release) using the Sequest algorithm (version 27) and the SORCERER (Sage-N Research) platform. Carbamidomethylation of cysteine was used as a fixed modification, and oxidation of methionine was used as a variable modification. In-gel digestion with trypsin (Thermo Fisher Scientific) and Coomassie blue staining were performed prior

to analysis by mass spectrometry. A protein database search of the biologically relevant experiments identified a total of 144 statistically significant proteins (95% peptide thresholds; 99% protein thresholds) in the sFasL–Fc-incubated samples compared to the Fc protein-incubated samples. Among these, 29 extracellular-space or plasma-membrane proteins were selected and are listed in *Table 1*. Although other proteins could potentially bind to sFasL, we selected TNFRSF10B (DR5), a TNF-receptor superfamily protein as the most likely candidate.

## Surface Plasmon resonance

A Biacore T100 system (GE Healthcare Bio-Sciences, Chicago, IL), installed at the National Center for Inter-University Research Facilities at Seoul National University, was used to perform surface plasmon resonance experiment. To determine equilibrium affinity measurements, recombinant DR5 (R and D systems: 631-T2-100) was coupled with a CM5 sensor chip using 10 mM sodium acetate (pH 5.5), and recombinant FasL or TRAIL (R and D Systems: 126-FL-010) was passed over the chip at a flow rate of 30 µL/min for 120 s. Dissociation was performed by adding 0.5 M NaOH at a flow rate of 30 µL/min for 240 s. A final response was calculated by subtracting the reading from an empty flow cell. BIA evaluation software version 3.1 (GE Healthcare) was used for to analyze the data. With exceptions of recombinant DR5, TRAIL, and FasL, all materials were purchased from GE Healthcare.

## Cell lines

Mouse EL4 and human Jurkat cells were purchased from the Korean Cell Line Bank (EL4 cells, 40039; Jurkat cells, 40152). The cell lines were authenticated by STR profiling and were negative for mycoplasma contamination.

## Plasmids and transfection

WT human *FAS* (NM_000043.5), *TNFRSF10B* (NM_147187.2), *FASLG* (AY858799.1), *TNF* (NC_000006.12), and *TNFSF10* (NM_001190942.2), and the mutant forms of human *TNFRSF10B* and *FASLG* (as described in *Figure 2—figure supplement 1B*) were cloned into the pIRES3-puro vector (TaKaRa Bio, Inc, Shiga, Japan). EL4 cells were transfected with these genes using a Neon Transfection System kit (once at 1080 V, 50 ms; Thermo Fisher Scientific), in accordance with the manufacturer's instructions. The expression of transfected genes was confirmed by real-time PCR (*Figure 1—figure supplement 4I*) or flow cytometry (*Figure 1—figure supplement 4K* and *Figure 2—figure supplement 1D and E*).

## Transfection of small-interfering RNA (siRNA)

Human siRNAs for *TNFRSF10A* (SASI_Hs01_00139573), *TNFRSF10B* (SASI_Hs01_00040567), *TNFRSF1A* (SASI_Hs01_00033456), *TNFRSF12A* (SASI_Hs01_00129286), *FAS* (SASI_Hs02_00301734), *RELA* (SASI_Hs01_00171091), *CUHK* (SASI_Hs01_00206921), and *IKBKB* (SASI_Hs01_00156170), and the MISSION siRNA universal negative control were purchased from Sigma-Aldrich (St. Louis, MO). Human synovial cells were transfected with siRNA by electroporation using the Neon Transfection System kit (Thermo Fisher Scientific), in accordance with the manufacturer's instructions. After 24 hr of transfection, cells were stimulated with 400 ng/mL sFasL (R and D Systems).

## CRISPR/Cas9 genome editing

TrueGuide sgRNA (*FAS*, no. A35511; CRISPR872691_SGM – GATCCAGATCTAACTTGGGG and *TNFRSF10B*; CRISPR606464_SGM – ACAACGAGCACAAGGGTCTT, and control, no. A35526) and Truecut Cas9 protein (no. A36498) were purchased from Thermo Fisher Scientific. In total, $5 \times 10^4$ hFLSCs were mixed with 7.5 pmol sgRNA, 7.5 pmol Cas9 protein, and 10 µL resuspension buffer R and electroporated using the Neon Transfection System kit (twice at 880 V for 35 ms). Transfected cells were cultured for 48 hr after electroporation. The transfection efficiencies of DR5 and Fas genes were >80%, and DR5- or Fas-negative cells were sorted using a fluorescence-activate cell-sorting Aria III instrument (BD Biosciences, San Diego, CA).

## Immunoprecipitation

For immunoprecipitation, $10^6$ hFLSCs per sample were incubated with 200 ng/mL His-tagged TNF-α, FasL, or TRAIL (R and D Systems) for 30 min at 37°C, followed by cross-linking with 5 mM BS³ and

20 mM quench solution (pH 7.5). Then, the cells were washed twice with cold PBS and lysed using ProteoPrep membrane extraction kit (Sigma-Aldrich) in the presence of protease and phosphatase inhibitor cocktails. To prepare denatured proteins for immunoprecipation, cells were lysed with buffer containing 50 mM of Tris (pH 7.9), 10 mM of dithiothreitol, 5 mM of EDTA, and 1% of SDS and then boiled for 10 min. The cell lysates (500 µg) were diluted with six volumes of non-denaturing cell lysis buffer and incubated with anti-His or anti-DR5 antibodies (1:50; Cell Signaling Technology, Inc, Beverly, MA), as well as isotype control rabbit mAb IgG (DA1E; Cell Signaling Technology, Inc) for 1 hr at 4°C. Samples were treated with 40 µL protein A/G PLUS-agarose beads (Santa Cruz Bio-technology, Santa Cruz, CA) overnight at 4°C, separated using 10% SDS-PAGE and transferred to a polyvinylidene fluoride membranes (Millipore, Billerica, MA) for western blotting.

## Immunoblotting

HFLSCs were stimulated with 200 ng/mL recombinant human sFasL (R and D Systems) in the presence or absence of anti-human DR5 and/or anti-human Fas blocking antibodies. The cells were washed twice with PBS after stimulation and lysed with buffer containing 20 mM Tris–Cl (pH 7.9), 120 mM NaCl, 0.5% Triton X–100, 2.5 mM EDTA, and 2 mM dithiothreitol in the presence of protease and phosphatase inhibitor cocktails. The eluted samples were separated using 8% polyacryl-amide SDS-PAGE and transferred to polyvinylidene fluoride membranes (Millipore) for western blotting. Antibodies for phospho-p65 (93H1), p65 (D14E12), phospho-IκBα (14D4), IκBα (L35A5), and β-actin (13E5), DR5 (D4E9), Fas (C18C12), His-Tag (D3I1O), TNF-α (D5G9), TRAIL (C92B9), and FasL (D1N5E) were purchased from (Cell Signaling Technology, Inc).

## Flow cytometry

Flow cytometry staining and analyses were performed according to standard protocols. The antibodies used are listed in Appendix 1—Key resource table. To measure cell death, phycoerythrin (PE)-conjugated AnnexinV (BD Biosciences) and 7-aminoactinomycin D (BD Biosciences) were used in accordance with the manufacturers' instructions. To monitor necroptosis, hFLSCs were treated with 50 µM Z–VAD–FMK followed by staining with propidium iodide (BD Biosciences). Binding of FasL to hFLSCs was assessed using FasL–Fc fusion proteins (Y Biologics, Seoul, Republic of Korea) and PE- or fluorescein isothiocyanate (FITC)-conjugated anti-human IgG Fc, or biotinylated FasL, TRAIL, and TNF-α, as well as streptavidin PE (SAv-PE; BioLegend). All in vitro staining for flow cytometry was performed in PBS containing 1% FBS and 0.09% sodium azide for 30 min at 4°C. FasL, TRAIL, and TNF-α proteins were biotinylated for 30 min at room temperature using 10 mM Sulfo-NHS-SS-Biotin (Thermo Fisher Scientific). Data were acquired using an LSR II flow cytometer and analyzed using FlowJo software (ver 10.0; Tree Star, Ashland, OR).

## Microarray assay

RNA was isolated from the joint tissues of WT, *Fas*<sup>lpr/lpr</sup>, and *Fasl*<sup>gld/gld</sup> mice 10 days after K/BxN serum transfer using the RNeasy Mini kit (Qiagen). To generate biotinylated-cRNA, total RNA was amplified and purified using an Ambion Illumina RNA amplification kit (Thermo Fisher Scientific), in accordance with the manufacturer's instructions. The quality of hybridization and overall chip perfor-mance were monitored by an internal quality control. Raw data were extracted using Illumina GenomeStudio software (Gene Expression Module v1.9.0; Illumina, Inc, San Diego, CA). We applied a filtering criterion for data analysis and a high signal value was required to obtain a detection p-value <0.05. Selected gene signal values were transformed logarithmically, and statistical signifi-cance of the expression data was determined using fold change. Gene enrichment and functional annotation analysis for the significant probe list were performed using DAVID (http://david.abcc.ncifcrf.gov/home.jsp). All data analyses and visualization of differentially expressed genes were per-formed using R software (ver. 2.14.0; R Development Core Team, Vienna, Austria). The data were deposited at GEO (accession no. GSE110343).

## ELISA

ELISA kits (BD Biosciences and R and D Systems) were used to measure human CCL2, CCL3, and CXCL10 proteins and human and mouse CX3CL1. All assays were performed in accordance with the

manufacturer's instructions. Standard curves were generated using recombinant proteins provided by the manufacturers.

### Real-time PCR analyses

Total RNA was isolated from mouse joint tissues or human synovial cells using an RNeasy kit (Qiagen), in accordance with the manufacturer's instructions. Joint tissues were prepared at 10 days after K/BxN serum transfer as described previously (*Kim and Chung, 2012*). RNA was reverse transcribed into cDNA using M-MLV-RT reverse transcriptase (Promega Corp., Madison, WI) prior to quantitative real-time PCR analyses. Gene-specific PCR products were measured using a 7500 sequence detection system (Applied Biosystems, Inc, Foster City, CA), and the results for each cytokine were normalized against Gapdh expression. All primers and probes were synthesized by Applied Biosystems (Appendix 1—Key resource table) and were used with a SensiFAST Probe Lo-ROX One-Step Kit (Bioline, London, UK).

### Migration assay

Migration was measured using 8 μm pore size inserts (BD Biosciences) in accordance with the manufacturer's instructions. Briefly, 24-well plates were coated with Matrigel (extracellular matrix, Sigma-Aldrich) and the inserts placed into RPMI 1640 media in the presence or absence of sFasL (400 ng/mL or indicated concentrations; R and D Systems) or CX3CL1 (400 ng/mL or indicated concentrations; R and D Systems). The cells were cultured on the inserts and incubated for 24 hr at 37˚C. Cell migration was measured by counting the number of cells in the bottom of each chamber, compared with the number of migrated cells in the absence of stimulant. Anti-mouse CX3CR1 monoclonal antibodies (QA16A03; Biolegend) were added at 1 hr before stimulation to block the CX3CL1–CX3CR1 interaction.

### Statistical analyses

Data were analyzed using GraphPad Prism software (ver. 5.0; GraphPad Software Inc, San Diego, CA), and t-tests were performed to compare two groups. One-way analysis of variance (ANOVA) and Tukey's post hoc tests were used to compare groups. A p-value < 0.05 was considered statistically significant.

## Acknowledgements

We would like to thank Hyehwa Forum members for their helpful discussions. This research was supported by Korea Health Technology R and D Project grant through the Korea Health Industry Development Institute (KHIDI), which is funded by the Ministry of Health and Welfare, Republic of Korea (grant number: HI14C1277).

## Additional information

### Funding

| Funder | Grant reference number | Author |
| --- | --- | --- |
| Ministry of Health and Welfare | HI14C1277 | Doo Hyun Chung |

The funders had no role in study design, data collection and interpretation, or the decision to submit the work for publication.

### Author contributions

Dongjin Jeong, Data curation, Formal analysis, Validation, Investigation, Visualization, Methodology; Hye Sung Kim, Hye Young Kim, Conceptualization, Investigation, Methodology; Min Jueng Kang, Hyeryeon Jung, Yumi Oh, Donghyun Kim, Jaemoon Koh, Sung-Yup Cho, Ho Min Kim, Investigation, Methodology; Yoon Kyung Jeon, Investigation; Eun Bong Lee, Resources; Seung Hyo Lee, Eui-Cheol Shin, Interpretation and integration of key experiments and results; Eugene C Yi, Methodology; Doo

Hyun Chung, Conceptualization, Data curation, Formal analysis, Supervision, Funding acquisition, Validation, Investigation, Methodology, Project administration

### Author ORCIDs

Dongjin Jeong https://orcid.org/0000-0001-7122-1220
Hye Sung Kim https://orcid.org/0000-0003-1297-7081
Hye Young Kim https://orcid.org/0000-0001-5978-512X
Hyeryeon Jung https://orcid.org/0000-0001-5179-8604
Donghyun Kim https://orcid.org/0000-0001-7863-7384
Yoon Kyung Jeon http://orcid.org/0000-0001-8466-9681
Eun Bong Lee https://orcid.org/0000-0003-0703-1208
Eui-Cheol Shin https://orcid.org/0000-0002-6308-9503
Ho Min Kim https://orcid.org/0000-0003-3029-3643
Doo Hyun Chung https://orcid.org/0000-0002-9948-8485

### Ethics

Human subjects: Research participants were diagnosed with rheumatoid arthritis, fulfilled the 1987 American College of Rheumatology (ACR) criteria. There were no specific characteristics within the participants other than rheumatoid arthritis. All participants with arthritis were recruited in Seoul National University Hospital. This study was approved by the Institutional Review Board of Seoul National University Hospital (H 1009-064-332) and informed consent obtained from all sample donors.

Animal experimentation: This study was approved by the Institutional Animal Care and Use Committee of Biomedical Research Institute of Seoul National University Hospital (BRISNUH, IACUC No. 17-0051 and 20-0171). All of the animals were maintained in the facility accredited AAALAC International (#001169) in accordance with Guide for the Care and Use of Laboratory Animals 8th edition, NRC (2010). All animal experiments were done according to the Guideline for Ethical Animal Experiments, Seoul National University (2013). Mouse sacrifice was performed under isoflurane anesthesia, and every effort was made to minimize suffering.

### Decision letter and Author response

Decision letter https://doi.org/10.7554/eLife.48840.sa1
Author response https://doi.org/10.7554/eLife.48840.sa2

## Additional files

### Supplementary files

• Transparent reporting form

### Data availability

The microarray data mentioned in this paper has been deposited in NCBI's Gene Expression Omnibus (GEO) and are accessible through GEO Series accession number GSE110343.

The following dataset was generated:

| Author(s) | Year | Dataset title | Dataset URL | Database and Identifier |
|---|---|---|---|---|
| Kim HS, Chung DH | 2018 | Genome-wide analysis for joint tissues of Fas (ligand) mutant mice during autoantibody induced arthritis | https://www.ncbi.nlm.nih.gov/geo/query/acc.cgi?&acc=GSE110343 | NCBI Gene Expression Omnibus, GSE110343 |

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

# Appendix 1

**Appendix 1—key resources table**

| Reagent type (species) or resource | Designation | Source or reference | Identifiers | Additional information |
|---|---|---|---|---|
| Antibody | anti-DR5 antibody (Polyclonal) | Abcam | Abcam:ab-8416; RRID:AB_306551 | Immunohistochemistry (1:50, v/v) |
| Antibody | PE-conjugated Annexin V | BD Biosciences | BD:556421 | Flow cytometry(1:50, v/v) |
| Antibody | Alexa700-conjugated anti-mouse CX3CR1(SA011F11) | Biolegend | Biolegend:149035; RRID:AB_2629605 | Flow cytometry(1:200, v/v) |
| Antibody | APC-conjugated anti-human Fas(DX2) | Biolegend | Biolegend:305612; RRID:AB_314550 | Flow cytometry(1:200, v/v) |
| Antibody | APC-conjugated anti-mouse CX3CR1(SA011F11) | Biolegend | Biolegend:149007; RRID:AB_2564491 | MACS-sorting (1:50, v/v) |
| Antibody | APC-conjugated anti-mouse F4/80(BM8) | Biolegend | Biolegend:123115; RRID:AB_893493 | Flow cytometry(1:200, v/v) |
| Antibody | BV421-conjugated anti-human FasL(NOK-1) | Biolegend | Biolegend:306411; RRID:AB_2716104 | Flow cytometry(1:200, v/v) |
| Antibody | FITC-conjugated anti-human IgG Fc (HP6017) | Biolegend | Biolegend:409309; RRID:AB_2561854 | Flow cytometry(1:200, v/v) |
| Antibody | FITC-conjugated anti-mouse CD11b(M1/70) | Biolegend | Biolegend:101205; RRID:AB_312788 | Flow cytometry(1:200, v/v) |
| Antibody | PE-conjugated anti-human DR5(DJR2-4) | Biolegend | Biolegend:307405; RRID:AB_314677 | Flow cytometry(1:200, v/v) |
| Antibody | PE-conjugated anti-human IgG Fc (HP6017) | Biolegend | Biolegend:409303; RRID:AB_10900424 | Flow cytometry(1:200, v/v) |
| Antibody | PE-conjugated anti-mouse DR5(MD5-1) | Biolegend | Biolegend:119905; RRID:AB_345401 | Flow cytometry(1:200, v/v) |
| Antibody | PE-Cy7-conjugated anti-mouse Ly6G(1A8) | Biolegend | Biolegend:127617; RRID:AB_1877262 | Flow cytometry(1:200, v/v) |
| Antibody | PerCP-Cy5.5-conjugated anti-mouse CD45(30-F11) | Biolegend | Biolegend:103131; RRID:AB_893344 | Flow cytometry(1:200, v/v) |
| Antibody | Ultra-LEAF Purified anti-mouse CX3CR1 Recombinant antibody(QA16A03) | Biolegend | Biolegend:153707, RRID:AB_2721771 | In vitro treatment(10 μg/mL) |
| Antibody | DR5 (D4E9) XP Rabbit mAb (D4E9) | Cell signaling | Cell signaling:8074T; RRID:AB_10950817 | Western blot, Co-IP (1:1000, v/v, 1:50, v/v (Co-IP)) |
| Antibody | Fas (C18C12) Rabbit mAb (C18C12) | Cell signaling | Cell signaling:4233T; RRID:AB_2100359 | Western blot(1:1000, v/v) |
| Antibody | FasL (D1N5E) Rabbit mAb (D1N5E) | Cell signaling | Cell signaling:68405S; RRID:AB_2799745 | Western blot(1:1000, v/v) |
| Antibody | His-Tag (D3I1O) XP Rabbit mAb(D3I1O) | Cell signaling | Cell signaling:12698S; RRID:AB_2744546 | Co-IP (1:1000, v/v) |
| Antibody | IκBα (L35A5) Mouse mAb | Cell signaling | Cell signaling:4814; RRID:AB_390781 | Western blot (1:1000, v/v) |
| Antibody | NF-κB p65 (D14E12) XP Rabbit mAb | Cell signaling | Cell signaling:8242; RRID:AB_10859369 | Western blot (1:1000, v/v) |

*Continued on next page*

*Appendix 1—key resources table continued*

| Reagent type (species) or resource | Designation | Source or reference | Identifiers | Additional information |
|---|---|---|---|---|
| Antibody | Phospho-IκBα (Ser32) (14D4) Rabbit mAb | Cell signaling | Cell signaling:2859; RRID:AB_561111 | Western blot (1:1000, v/v) |
| Antibody | Phospho-NF-κB p65 (Ser536) (93H1) Rabbit mAb | Cell signaling | Cell signaling:3033; RRID:AB_331284 | Western blot (1:1000, v/v) |
| Antibody | TRAIL (C92B9) Rabbit mAb (C92B9) | Cell signaling | Cell signaling:3219S; RRID:AB_2205818 | Western blot (1:1000, v/v) |
| Antibody | β-Actin (13E5) Rabbit mAb | Cell signaling | Cell signaling:4970; RRID:AB_2223172 | Western blot (1:1000, v/v) |
| Antibody | Anti-Mouse CD178 (Fas Ligand)(MFL3) | eBioscience | eBioscience:16-5911; RRID:AB_469145 | In vivo injection (50 μg / injection) |
| Antibody | Anti-Fas Antibody (human, neutralizing)(ZB4) | Merck | Merck:05-338; RRID:AB_309682 | In vitro treatment (20 μg/mL) |
| Antibody | Human Fas Ligand/TNFSF6 Antibody(100419) | R&D | R&D:MAB126; RRID:AB_2246667 | Neutralization (10 μg/mL) |
| Antibody | Human TRAIL R2/TNFRSF 10B Antibody(Polyclonal) | R&D | R&D:AF631; RRID:AB_355489 | Neutralization (20 μg/mL) |
| Antibody | Human/Mouse CX3CR1 Antibody (Polyclonal) | R&D | R&D:AF5825; RRID:AB_2292441 | In vivo injection (10 μg / injection) |
| Antibody | Mouse Fas/TNFRSF6/CD95 Antibody (Polyclonal) | R&D | R&D:AF435; RRID:AB_355358 | In vivo injection (50 μg / injection) |
| Antibody | Mouse IgG2B Isotype Control(73009) | R&D | R&D:MAB0042; RRID:AB_471245 | Control (10 μg/mL) |
| Antibody | Mouse TRAIL R2/TNFRSF 10B Antibody(Polyclonal) | R&D | R&D:AF721; RRID:AB_205069 | In vivo injection (50 μg / injection) |
| Antibody | Normal Goat IgG Control (Polyclonal) | R&D | R&D:AB-108-C; RRID:AB_354267 | Control (20 μg/mL) |
| Antibody | Normal Goat IgG Control (Polyclonal) | R&D | R&D:AB-108-C; RRID:AB_354267 | In vivo injection (50 μg / injection) |
| Biological sample (*H. sapiens*) | Human fibroblast-like synoviocytes (hFLSCs) | PMID:19709444 | | See materials and methods |
| Cell line (*M. musculus*) | EL4 | Korean Cell Line Bank | KCLB:40039 | |
| Cell line (*M. musculus*) | Jurkat | Korean Cell Line Bank | KCLB:40152 | |
| Chemical compound, drug | Z-VAD-FMK | Calbiochem | Calbiochem:187389-52-2 | 50 μM |
| Chemical compound, drug | Z-VAD-FMK | Calbiochem | Calbiochem:187389-52-2 | 50 μg / mouse |
| Chemical compound, drug | Protein A/G PLUS-agarose beads | Santa Cruz Biotechnology | Sanat cruz:sc-2003 | |
| Chemical compound, drug | BMS345541 | Sigma | Sigma:B9935 | 10 mM |
| Chemical compound, drug | MG132 | Sigma | Sigma:M7449 | 10 mM |
| Chemical compound, drug | NSCI | Sigma | Sigma:N1413 | 5 μM |
| Chemical compound, drug | PD980259 | Sigma | Sigma:513000 | 10 mM |
| Chemical compound, drug | SB203580 | Sigma | Sigma:S8307 | 10 mM |

*Continued on next page*

*Appendix 1—key resources table continued*

| Reagent type (species) or resource | Designation | Source or reference | Identifiers | Additional information |
|---|---|---|---|---|
| Chemical compound, drug | U0126 | Sigma | Sigma:U120 | 10 mM |
| Chemical compound, drug | BS3 (bis-(sulfo-succinimidyl) suberate) | Thermo Fisher Scientific | Thermo Fisher:21580 | See materials and methods |
| Chemical compound, drug | EZ Link Sulfo-NHS-SS-Biotin | Thermo Fisher Scientific | Thermo Fisher: A39258 | Biotinylation of Fas ligand (10mM) |
| Commercial assay or kit | Anti-APC microbeads | Miltenyi Biotec | Miltenyi Biotec:130-090-855 | Cell isolation (1:50, v/v) |
| Commercial assay or kit | RNeasy Mini kit | Qiagen | Qiagen:74104 | |
| Commercial assay or kit | Human CCL2/MCP-1 Quantikine ELISA Kit | R&D | R&D:SCP00 | |
| Commercial assay or kit | Human CCL3/MIP-1 alpha Quantikine ELISA Kit | R&D | R&D:SMA00 | |
| Commercial assay or kit | Human CX3CL1/Fractalkine DuoSet ELISA | R&D | R&D:DY365 | |
| Commercial assay or kit | Human CXCL10/IP-10 Quantikine ELISA Kit | R&D | R&D:SIP100 | |
| Commercial assay or kit | Mouse CX3CL1/Fractalkine DuoSet ELISA | R&D | R&D:DY472 | |
| Commercial assay or kit | Mouse Fas Ligand/TNFSF6 Quantikine ELISA Kit | R&D | R&D;MFL00 | |
| Genetic reagent (*M. musculus*) | Taqman gene expression assay (*Ccl2*) | Thermo Fisher Scientific | Thermo Fisher: Mm00441242_m1 | |
| Genetic reagent (*M. musculus*) | Taqman gene expression assay (*Ccl3*) | Thermo Fisher Scientific | Thermo Fisher: Mm00441259_g1 | |
| Genetic reagent (*H. sapiens*) | Taqman gene expression assay (*CX3CL1*) | Thermo Fisher Scientific | Thermo Fisher: Hs00171086_m1 | |
| Genetic reagent (*M. musculus*) | Taqman gene expression assay (*Cx3cl1*) | Thermo Fisher Scientific | Thermo Fisher: Mm00436454_m1 | |
| Genetic reagent (*M. musculus*) | Taqman gene expression assay (*Cxcl10*) | Thermo Fisher Scientific | Thermo Fisher: Mm00445235_m1 | |
| Genetic reagent (*H. sapiens*) | Taqman gene expression assay (*GAPDH*) | Thermo Fisher Scientific | Thermo Fisher: Hs02786624_g1 | |
| Genetic reagent (*M. musculus*) | Taqman gene expression assay (*Gapdh*) | Thermo Fisher Scientific | Thermo Fisher: Mm99999915_g1 | |
| Genetic reagent (*M. musculus*) | Taqman gene expression assay (*Il1b*) | Thermo Fisher Scientific | Thermo Fisher: Mm00434228_m1 | |
| Genetic reagent (*M. musculus*) | Taqman gene expression assay (*Il4*) | Thermo Fisher Scientific | Thermo Fisher: Mm00445259_m1 | |
| Genetic reagent (*M. musculus*) | Taqman gene expression assay (*Il6*) | Thermo Fisher Scientific | Thermo Fisher: Mm00446190_m1 | |
| Genetic reagent (*M. musculus*) | Taqman gene expression assay (*Tgfb*) | Thermo Fisher Scientific | Thermo Fisher: Mm01178820_m1 | |
| Genetic reagent (*M. musculus*) | Taqman gene expression assay (*Tnfa*) | Thermo Fisher Scientific | Thermo Fisher: Mm00443258_m1 | |
| Genetic reagent (*H. sapiens*) | Taqman gene expression assay (*TNFRSF10B*) | Thermo Fisher Scientific | Thermo Fisher: Hs00366278_m1 | |
| Genetic reagent (*H. sapiens*) | Taqman gene expression assay (*TNFRSF1A*) | Thermo Fisher Scientific | Thermo Fisher: Hs01042313_m1 | |

*Continued on next page*

*Appendix 1—key resources table continued*

| Reagent type (species) or resource | Designation | Source or reference | Identifiers | Additional information |
|---|---|---|---|---|
| Genetic reagent (*H. sapiens*) | Taqman gene expression assay (*TNFRSF10A*) | Thermo Fisher Scientific | Thermo Fisher: Hs00269492_m1 | |
| Genetic reagent (*H. sapiens*) | Taqman gene expression assay (*TNFRSF12*) | Thermo Fisher Scientific | Thermo Fisher: Hs00171993_m1 | |
| Genetic reagent (*M. musculus*) | Taqman gene expression assay (*Tnfrsf10b*) | Thermo Fisher Scientific | Thermo Fisher: Mm00457866_m1 | |
| Genetic reagent (*H. sapiens*) | TrueGuide sgRNA (*FAS*) | Thermo Fisher Scientific | Thermo Fisher: CRISPR872691_SGM | 7.5 pmol |
| Genetic reagent (*H. sapiens*) | TrueGuide sgRNA (negative control) | Thermo Fisher Scientific | Thermo Fisher: A35526 | 7.5 pmol |
| Genetic reagent (*H. sapiens*) | TrueGuide sgRNA (*TNFRSF10B*) | Thermo Fisher Scientific | Thermo Fisher: CRISPR606464_SGM | 7.5 pmol |
| Other | 7-AAD | BD Biosciences | BD:559925 | Flow cytometry(1:50, v/v) |
| Other | Propidium Iodide Staining Solution | BD Biosciences | BD:556463 | Flow cytometry(1:50, v/v) |
| Peptide, recombinant protein | Recombinant Human Fas Ligand/TNFSF6 Protein | R&D | R&D:126-FL | In vitro treatment(2 µM) |
| Peptide, recombinant protein | Recombinant Human TRAIL R2/TNFRSF10B Fc Chimera Protein | R&D | R&D:631-T2 | In vitro treatment(200 ng/mL) |
| Peptide, recombinant protein | Recombinant Human TRAIL/TNFSF10 Protein | R&D | R&D:375-TL | Flow cytometry(200 ng/mL) |
| Peptide, recombinant protein | Recombinant Mouse CX3CL1/Fractalkine (Full Length) Protein | R&D | R&D:472-FF | In vitro treatment(200 ng/mL) |
| Peptide, recombinant protein | Recombinant Mouse CX3CL1/Fractalkine (Full Length) Protein | R&D | R&D:472-FF | In vivo injection (1 µg/injection) |
| Peptide, recombinant protein | Recombinant Mouse Fas Ligand/TNFSF6 Protein | R&D | R&D:526-SA | In vivo injection(1 µg/mouse) |
| Peptide, recombinant protein | Recombinant Mouse Fas Ligand/TNFSF6 Protein | R&D | R&D:526-SA | In vitro treatment(200 ng/mL) |
| Peptide, recombinant protein | Recombinant Mouse TRAIL | R&D | R&D:1121-TL | In vitro treatment (200 ng/mL) |
| Peptide, recombinant protein | Recombinant Mouse TRAIL | R&D | R&D:1121-TL | In vivo injection (1 µg/mouse) |
| Peptide, recombinant protein | Truecut Cas9 protein | Thermo Fisher Scientific | Thermo Fisehr: A36498 | 7.5 pmol |
| Peptide, recombinant protein | FasL-Fc | Y-Biologics | This paper | See materials and methods |
| Peptide, recombinant protein | Recombinant Human TNF-alpha | R&D | R&D:210-TA | See materials and methods |
| Peptide, recombinant protein | Recombinant Mouse TNF-alpha | R&D | R&D:410-MT | See materials and methods |
| Recombinant DNA reagent | pIRESpuro3 | Clontech | Clontech:631619 | See materials and methods |
| Sequence-based reagent | MISSION siRNA (*CUHK*) | Sigma | Sigma:SASI_Hs01_00206921 | |
| Sequence-based reagent | MISSION siRNA (*FAS*) | Sigma | Sigma:SASI_Hs02_00301734 | |

*Continued on next page*

*Appendix 1—key resources table continued*

| Reagent type (species) or resource | Designation | Source or reference | Identifiers | Additional information |
|---|---|---|---|---|
| Sequence-based reagent | MISSION siRNA (*IKBKB*) | Sigma | Sigma:SASI_Hs01_00156170 | |
| Sequence-based reagent | MISSION siRNA (*RELA*) | Sigma | Sigma:SASI_Hs01_00171091 | |
| Sequence-based reagent | MISSION siRNA (*TNFRSF10A*) | Sigma | Sigma:SASI_Hs01_00139573 | |
| Sequence-based reagent | MISSION siRNA (*TNFRSF10B*) | Sigma | Sigma:SASI_Hs01_00040567 | |
| Sequence-based reagent | MISSION siRNA (*TNFRSF12A*) | Sigma | Sigma:SASI_Hs01_00129286 | |
| Sequence-based reagent | MISSION siRNA (*TNFRSF1A*) | Sigma | Sigma:SASI_Hs01_00033456 | |
| Software, algorithm | FlowJo | FlowJo | Version 10 | |
| Software, algorithm | BIA evaluation | GE Healthcare Bio-Sciences | GE:BR-1005-97 | |
| Software, algorithm | Illumina GenomeStudio | Gene Expression Module | Version 1.9.0 | |
| Software, algorithm | GraphPad Prism | GraphPad Software | Version 5.0 | |
| Software, algorithm | Sequest algorithm | N/A | Version 27 | |
| Software, algorithm | SORCERER | Sage-N Research | | |
| Strain, strain background (*Mus musculus*, C57BL/6J) | *Fasl*^gld/gld^ | Japan SLC | | |
| Strain, strain background (*Mus musculus*, C57BL/6J) | *Fas*^lpr/lpr^ | Japan SLC | | |
| Strain, strain background (*Mus musculus*, C57BL/6J) | K/BxN | PMID:8945509 | | |
| Strain, strain background (*Mus musculus*, C57BL/6J) | *Tnfrsf10b* KO | MMMRC | MMMRC:030532-MU | |
| Strain, strain background (*Mus musculus*, C57BL/6J) | *Fasl*^Δm/Δm^ | PMID:19794494 | | |
| Strain, strain background (*Mus musculus*, C57BL/6J) | *Fasl*^Δs/Δs^ | PMID:19794494 | | |
| Strain, strain background (*Mus musculus*, C57BL/6J) | Wild type C57BL/6 (WT) | Orient Bio | | |

*Continued on next page*

*Appendix 1—key resources table continued*

| Reagent type (species) or resource | Designation | Source or reference | Identifiers | Additional information |
|---|---|---|---|---|
| Strain, strain background (*Mus musculus*, C57BL/6J) | *Cx3cr1* KO | Taconic | Taconic:4167 | |
| Strain, strain background (*Mus musculus*, C57BL/6J) | *Fas*$^{-/-}$ | PMID:7581453 | RIKEN BRC: RBRC01474 | |

