## [Decision Letter]

**Acceptance summary:**

In this study, you show that Fas ligand functions, not only by binding Fas, but also by binding to DR5 – a receptor that until now has been known to be activated only by the ligand TRAIL. Moreover, in exploring the contribution of this novel interaction to the pathology of autoantibody inducing arthritis in the mice K/BxN serum-transfer model, you found that soluble Fas ligand can induce via DR5 a functional change – transcription of the cytokine CX3CL1 (Fractalkine) – that does not occur upon triggering DR5 function by TRAIL, nor can it be induced via Fas. The induction of CX3CL1, and the consequent induction of several cytokines by the latter, are shown in your study to make a major contribution to the joint pathology in the K/BxN serum transfer model. Overall, this paper makes an important contribution to our knowledge of the functions served by the TNF family in revealing a novel interaction between members of the TNF ligand and receptor families, with a functional consequence that seems to contribute to the pathology of autoimmune diseases.

**Decision letter after peer review:**

Thank you for submitting your article "DR5 is a Fas-independent receptor for soluble Fas ligand that promotes inflammation" for consideration by *eLife*. Your article has been reviewed by 3 peer reviewers, and the evaluation has been overseen by a Reviewing Editor and Tadatsugu Taniguchi as the Senior Editor. The following individual involved in review of your submission has agreed to reveal their identity: Marcus E Peter (Reviewer #2).

The reviewers have discussed the reviews with one another and the Reviewing Editor has drafted this decision to help you prepare a revised submission.

Your manuscript was examined by three expert reviewers and a reviewing editor. They were all highly intrigued by your claimed finding that soluble Fas ligand binds DR5 and activates it. They all consider this finding to be of major importance. However, since during the long time that has passed since the discovery of Fas ligand and DR5 none has reached similar findings and, moreover, some studies seemed to have clearly demonstrated that Fas ligand does not bind DR5 (e.g. Figure 2A in PMID 9373179), they were all also concerned that these findings might be mistaken.

Each of the reviewers sent me a long list of suggestions of ways by which you may address this concern. To assist you coping with these numerous requests I assembled them in three groups that are presented below: (i) Requests that might involve doing some additional experiments, which must be met in order to allow me considering your paper for publication. (ii) Requests concerning just your way of writing, which must also be met. (iii) Additional requests that I consider to be of secondary importance for the central message of the paper. You might choose either to address these requests or just to delete the data that they concern.

I would like to assist you as much as I can in publishing your interesting findings, However, I feel just as well compelled to do all that is necessary to exclude any mistake.

(i) Requests that might involve doing additional experiments, which must be met in order to allow me considering your paper for publication.

1. Much more detailed information should be provided in the Figure Legends and Methods sections about key results pertaining to the proposed interaction of FasL and DR5.

In particular:

(a) Detailed information of the antibodies that were used in vivo and ex vivo should be presented. What clone of monoclonal antibodies was used? What is the evidence for mouse- human cross-reactivity (in case of use of antibody against the one to block the other)? etc.

(b) The concentrations of all the recombinant proteins and antibodies used should be specified.

(c) The number of experiments performed (biological n) should be specified.

(d) The mass spectrometry experiment should be described in much greater detail, for example – for how long was sFasL applied, at what temperature, at what concentration, to how many cells? What does "Fc protein was used as a negative control throughout the process" mean? The list of hits obtained from the AP-MS experiment should be fully presented to allow examining if other TNFRSF members were also identified.

2. All commercially purchased reagents that have been used in key experiments, e.g. sFasL, FasL-Fc, TRAIL, anti-Fas, and anti-DR5 antibodies must be characterized and validated in order to confirm that their identity and activity are indeed what they are advertised to be.

3. The specific binding of sFasL to DR5 should be reconfirmed by demonstrating co-immunoprecipitation of the two in endogenous setting, that is to say – using cells that express DR5 at its endogenous levels and using sFaL at concentrations of the recombinant protein that mimic endogenous levels. Assessment of the bioactivity of the ligands should also be done at concentrations of the recombinant protein that are at the range of the endogenous levels.

4. Knockdown does not fully abolish the expression of a protein, nor can the use of antibodies always allow to fully block its activity. To alleviate the concern that such incomplete effects have led to mistaken conclusions the authors are requested to apply CRISPR/Cas9 to knockout the DR5 and Fas genes instead (which is rather easy to do nowadays). They should then further reconfirm the findings by re-expressing the cDNA for the knocked-out genes. Does the knockout of DR5 eliminate a residual response of Fas deficient cells to FasL? Does re-expression of DR5 in such cells rescue such deficiency?

5. The assessment of the affinity of binding of TRAIL to DR5 should be presented in a figure.

6. Specific triggering of cell death by DR5 in response to sFasL should be validated by the use of inhibitors that specifically affect apoptosis and necroptosis.

(ii) Requests concerning just your way of writing, which must also be met.

1. The authors begin the abstract by noting that "Fas-independent membrane bound receptor for FasL has not yet been reported". In the introduction, they argue that "in many studies demonstrating the biological effects of FasL in vitro and in vivo, some of the results have not been confirmed to be Fas-dependent. Thus, it is reasonable to consider the possibility that sFasL may bind Fas-independent receptor in addition to Fas in vivo". The premise that an alternate receptor for sFasL exists is arbitrary: just because this hasn't been ruled out doesn't qualify it as a valid possibility. There has to be some positive evidence in favor of this specific notion in order to qualify it as a starting point or hypothesis for further studies. Please change this statement accordingly.

2. Fas lpr mice are not completely negative for Fas. Some tissues actually express wt Fas (PMID: 7528670). That could compound some of the data especially the in vivo data. Please discuss the implications.

3. An interesting consideration is whether the reported increase in metastases in TRAIL-R k.o mice by the Walczak group could be due to FasL unbound to TRAIL-R driving metastasis through increased binding to Fas. In addition, other reports suggested that TRAIL-R deficiency sensitized mice to inflammation, DSS induced colitis or DEN induced liver cancer (PMIDs: 24117005; 18079962). Again, that seems to be at odds with the activity of DR5 described here to promote inflammation. Please discuss the implications.

4. Not once do the authors explain to a wider audience what FLSC stands for.

5. Figure 1G: How many patients/slides were stained? Statistics need to be provided.

6. Line 104: It should read "Figure 1—figure supplement 2C".

7. Line 107: It should read "Figure 1—figure supplement 2D and E". 8) Line 133: It should read "Figure 2—figure supplement 1B" instead of Figure S4B.

8. Figure 3—figure supplement 2 is labeled in the figure legend as Supplemental figure 7, Figure 5—figure supplement 1 as Supplemental figure 9, and Figure 6—figure supplement 1 as Supplemental figure 10, respectively.

9. The authors state that based on the data sFasL-DR5 interactions induce both apoptosis and necroptosis. Without more detailed analyses it is not clear whether necroptosis is induced. All that the data suggest is that cells undergo secondary necrosis, a phenomenon typical for in vitro stimulated cells. Please discuss.

10. As to the article title. I would suggest to include in the title that this is in the context of autoantibody-induced arthritis.

Also the "Fas independent receptor" does not sound good.

Something like:

Fas ligand drives autoantibody-induced arthritis through binding to DR5/TRAIL-R2

(iii) Additional requests that I consider to be of secondary importance for the central message of the paper. You might choose either to address these requests or just to delete the data that they concern.

1. FasL-Fc is an unnatural construct that (presumably) displays two FasL homotrimers. Rather than using it interchangeably without explanation, the authors should better use just sFasL throughout the study.

2. The authors contrast binding of FasL-Fc to human DR5 with human DR4. What about other TRAIL receptors, and other TNFRSF members? A more complete analysis is called for.

3. If FasL binds to DR5, then what specific binding site does it occupy? Are there any specific mutations that can disrupt this binding? Is the FasL binding site for DR5 related to TRAIL's? Is there homology between the DR5 and Fas binding sites for FasL?

4. All negative data using inhibitors (e.g. zVAD-fmk) need to be supported by positive control data for the inhibitors, particularly in the in vivo experiments.

5. The authors should provide evidence that sFasL leads to the assembly of a FADD-containing complex that is dependent of DR5 and not Fas using both murine and human Fas and DR5 KO cells.

6. sFas and TRAIL are known inducers of cell death, however, in the present manuscript the authors do not take into account this variable and the possible effect that cell death might have in, as an example, selecting for a particular cell population. The in vitro experiments should be validated in the presence of cell death inhibitors. In this regard, the authors mention different modes of cell death without testing for them. If such conclusions are to be drawn than the authors should use specific inhibitors of the difference cell death modalities.

7. Throughout the manuscript there are several controls missing. While the authors demonstrate the efficiency of deletion of the siRNAs used, it is imperative to assess the abolishment of the signaling that depends on that receptor in the presence of its respective bona fide ligands. This can be achieved by assessing of the formation of the FADD-containing complex by western blot and downstream output of cell death and gene activation.

8. The western blot presented in figure 3j raises doubt regarding the efficiency of Fas blocking in their experiments. As demonstrated in lanes 7-9 versus lanes 1-3, their blocking antibody for Fas has very little capacity to block sFasL-mediated p-IkBα. The authors should explain this observation and (as specified above) repeat the experiment using KO cells for DR5 and/or Fas.

9. Regarding the mechanisms of AIA and its dependency on DR5-sFasL engagement the authors should show the levels of CX3CL1 in mFasL KO mice and in sFasKO mice treated with recombinant sFasL. Additionally, the authors should generate DR5/CX3CL DKO mice and DR5-FasL delt.s DKO to show that these mice do not exhibit AIA upon sFasL treatment.

10. To show that sFasL-DR5 indeed leads to CX3CL1-induced AIA the authors should add to figure 6 a rescue experiment using sFasL-KO mice and conditionally induce CX3CL1 expression.

[Editors' note: further revisions were suggested prior to acceptance, as described below.]

Thank you for submitting the revised version of your manuscript "Soluble Fas ligand drives autoantibody-induced arthritis via binding to DR5/TRAIL-R2" for consideration by *eLife*.

The manuscript has been reviewed by 3 peer reviewers, and the evaluation has been overseen by a Reviewing Editor and Tadatsugu Taniguchi as the Senior Editor. They were impressed by the added information, but still had specific concerns and requests that must be addressed before a decision on your manuscript can be taken.

Comments about the ex vivo data:

Specific comments

1. The viewers believe that in order to exclude remaining doubts about your claims there is the need for several more rigorous controls, as detailed below:

Figure 1H

The figure shows binding of FasL-Fc to DR5-transfected EL4 cells, which is blocked by anti-FasL antibody.

There should be a comparison to binding of TRAIL-Fc and to an irrelevant Fc fusion protein. Also, does TRAIL-Fc or anti-DR5 antibody block this FACS shift observed with FasL-Fc? The effect of re-expression of Fas alone, without the re-expression of DR5, in the double knockout cells should also be shown. Similar questions apply to Figure 1I.

Figure 1I and Figure 1—figure supplement 4C and 4D

The authors should provide real-time PCR evidence that the Fas and TRAIL genes have indeed been knocked out in all the cells in the preparations used in these experiments.

In Fig, 1J

sFasL or sTRAIL is added to hFLSCs, cross-linked, and immunoprecipitated with IgG or anti-DR5 and FasL or TRAIL visualized by immunoblot. PBS is an inadequate control here: they should use another soluble TNFSF ligand as a negative control. Also, the binding of sFasL should be blocked by excess sTRAIL and vice versa.

Figure 1K

Please compare binding of DR5-Fc with Fas-Fc. Compare binding to EL4-FasL with binding to EL4-TRAIL (positive) and EL4-mTNFa (negative). Is binding of DR5-Fc blocked by TRAIL or by anti-DR5 antibody?

2. Figure 3H and I and the related supplementary figures

The relevance of the signalling mechanism by which sCX3CL1 is induced to the rest of the paper is not clear. Moreover, given selectivity caveats with the small molecule the significance of the evidence about this issue is limited. The authors should either delete this part or relate to its implications as just putative.

General comments

1. How were transfected cell lines generated and validated? Please explain and present the data.

2. The references cited for TRAIL-DR5 interaction and co-crystal structures are incomplete.

Comments about the in vivo data:

As the authors themselves show (Figure 1—figure supplemental 1A), cells of the lpr mice still express some Fas, although at a greatly reduced level. This, unfortunately, casts some doubt on the interpretation of the in vivo data in this study. The authors should either fully delete all these data or present their implication as just putative. It is a pity that the authors have not used the Fas full knockout mice (which are available from Jackson laboratories). Demonstrating an effect of injected sFasL on those mice would have yielded unequivocal evidence.

Comments about the English:

The sentence ""the interaction of Fas-independent membrane bound receptor and FasL remains elusive or unclear" makes no sense. It implies the existence of previous evidence of such an alternative receptor. There is no such evidence and hence the starting point of the study is still poorly justified. One could change the sentence in the introduction to "whether sFasL-mediated inflammation is regulated via interaction with Fas in vivo has not been demonstrated directly". Also, the next sentence would make more sense if it started with: "By addressing these questions we identified….".

In the abstract the authors claim the following: "Affinity purification mass spectrometry analysis using human fibroblast-like synovial cells (hFLSCs) revealed DR5 as a FasL receptor." This statement is somewhat misleading. As the authors explain in their reply to the reviewers' comments, the mass-spectrometric analysis of the affinity precipitate they obtained with recombinant FasL-Fc, resulted in the identification of many different proteins as putative interactors with DR5 being one of many. It is therefore not correct to claim that the interaction with DR5 would be specific. In the response to the reviewers' comments the authors state that they "selected DR5" from a list of 29 proteins they found in the IP which contained a transmembrane domain or were expressed as extracellular proteins. Whereas this puts the starting point of the study into question (why did they not pick CD70, another TNFRSF member which they found in the list?), I understand the argument but in view of the reviewers they should stay closer to the data and relate to their findings differently. Something along the lines of the following would probably be more appropriate: "Affinity purification mass spectrometry analysis using human fibroblast-like synovial cells (hFLSCs) identified DR5 as one of several candidates as the elusive Fas-independent FasL receptor. Subsequent cellular and biochemical analysis revealed that DR5 can specifically interact with recombinant FasL-Fc protein, albeit with an approximately 60-fold lower affinity than Fas."

[Editors' note: further revisions were suggested prior to acceptance, as described below.]

Thank you for sending your article entitled "Soluble Fas ligand drives autoantibody-induced arthritis via binding to DR5/TRAIL-R2" for peer review at *eLife*. Your article is being evaluated by 4 peer reviewers, and the evaluation is being overseen by a Reviewing Editor and Tadatsugu Taniguchi as the Senior Editor.

Although you have addressed most of the concerns of the reviewers, the reviewers remained concerned about the lack of a definitive control from the beginning of this review process. This concern has not been addressed yet thoroughly enough. The reviewers maintain their request to provide such control by the use of a full Fas knockout mouse (which are available from Jackson Labs). Therefore, we ask you to obtain these mice and perform the definitive experiment which is to test whether injection of recombinant sFasL in full Fas knockout mice promotes AIA or not.

Given that this study may have far-reaching consequences, we believe that the revision with the above experimental results is essential for us to make the final decision.

One additional point: There seems to be a mistake in Figure 1H. The EL4 thymoma cells used in the experiment that was presented in this figure have been shown in quite a number of other studies to express TNF receptors and to respond to TNF. We wonder if the fact that you did not see any sign for binding of biotinylated TNF to the EL4-hDR5 cells that were used in this experiment was due to some fault in the preparation of the biotinylated TNF.

---

## [Author Response]

Your manuscript was examined by three expert reviewers and a reviewing editor. They were all highly intrigued by your claimed finding that soluble Fas ligand binds DR5 and activates it. They all consider this finding to be of major importance. However, since during the long time that has passed since the discovery of Fas ligand and DR5 none has reached similar findings and, moreover, some studies seemed to have clearly demonstrated that Fas ligand does not bind DR5 (e.g. Figure 2A in PMID 9373179), they were all also concerned that these findings might be mistaken.Each of the reviewers sent me a long list of suggestions of ways by which you may address this concern. To assist you coping with these numerous requests I assembled them in three groups that are presented below: (i) Requests that might involve doing some additional experiments, which must be met in order to allow me considering your paper for publication. (ii) Requests concerning just your way of writing, which must also be met. (iii) Additional requests that I consider to be of secondary importance for the central message of the paper. You might choose either to address these requests or just to delete the data that they concern.I would like to assist you as much as I can in publishing your interesting findings, However, I feel just as well compelled to do all that is necessary to exclude any mistake. eLife usually allows a maximal period of just two months for revision. I will make an exception in this case and will allow you resubmitting within three months from now. If addressing the requests that we are making will take you longer time, I will also be glad to consider the paper for publication. However, in that case the date of the original submission will be ignored.(i) Requests that might involve doing additional experiments, which must be met in order to allow me considering your paper for publication.1. Much more detailed information should be provided in the Figure Legends and Methods sections about key results pertaining to the proposed interaction of FasL and DR5. In particular:(a) Detailed information of the antibodies that were used in vivo and ex vivo should be presented. What clone of monoclonal antibodies was used? What is the evidence for mouse- human cross-reactivity (in case of use of antibody against the one to block the other)? etc.

As reviewer suggested, we provide detailed information for the antibodies used in our experiments in key resources table. Moreover, we checked cross-reactivity of antibodies between human and mouse by performing functional tests. To address this issue, we stimulated human and mouse cells with human or mouse soluble FasL in the presence or absence of antibodies against human or mouse Fas or DR5, and measured amounts of CX3CL1 in the culture supernatants and apoptosis (Figure 1—figure supplement 3 A-C). Antibodies against human DR5 inhibited CX3CL1 production in human, but not mouse synovial fibroblasts, whereas antibodies against mouse DR5 suppressed CX3CL1 production in mouse, but not human synovial fibroblasts. In addition, antibodies against human DR5 and Fas, but not those against mouse DR5 and Fas, inhibited cell death in human Jurkat cells, whereas antibodies against mouse DR5 and Fas, but not those against human DR5 and Fas, suppressed cell death in mouse spleen T cells. These findings indicate that antibodies against Fas and DR5 do not have cross reactivity between human and mouse Fas and DR5. These data are presented in Figure 1 —figure supplement 3A-C and described in the section of Results as follows: Anti-DR5 and Fas antibodies do not show cross reactivity between human and mouse DR5 and Fas, respectively (Figure 1—figure supplement 3).

(b) The concentrations of all the recombinant proteins and antibodies used should be specified.

As reviewer suggested, we have described the concentrations of all recombinant proteins and antibodies used in the methods and figure legends of revised manuscript and key resources table.

(c) The number of experiments performed (biological n) should be specified.

As reviewer suggested, we provided the numbers of experiments performed in figure legends.

(d) The mass spectrometry experiment should be described in much greater detail, for example – for how long was sFasL applied, at what temperature, at what concentration, to how many cells? What does "Fc protein was used as a negative control throughout the process" mean? The list of hits obtained from the AP-MS experiment should be fully presented to allow examining if other TNFRSF members were also identified.

We would like to appreciate the reviewer’s reasonable comments. As we described in the Materials and methods section on page 20 and 21, we implemented the in-situ chemical cross-linking method (Mapping protein receptor-ligand interactions via in vivo chemical crosslinking, affinity purification, and differential mass spectrometry, *Methods,* 2012) for identification of Fas-independent FasL receptor. Following the procedure, 20 μg of biotinylated sFasL-Fc protein was incubated with 1×10^8^ cells for 2 hours. After incubation, 1 mL of 3 μg/ml of BS^3^ (bis- (sulfo-succinimidyl) suberate) cross-linking reagent was added for 30 min at room temperature. Then, we quenched the cross-linking reaction by adding Tris-bis HCl (2 M, pH 7.4) to a final concentration of 15 mM and incubated at room temperature for 15 min. Cells were lysed in RIPA buffer and biotinylated-sFasL-Fc chemical cross-linked protein complexes were purified by avidin affinity chromatography. As a negative control, we used Fc protein to remove non-specific binding to Fc region which included in our protein of interest, sFasL-Fc protein. In this experiment, all other steps were treated identically to sFasL-Fc protein and Fc protein incubated samples. The purified proteins were separated by 4-12% SDS-PAGE, stained with Coomassie Blue, and digested in-gel with trypsin for mass spectrometry analysis. Protein database search of the biologically duplicate experiments resulted in a total of 144 proteins appeared statistically significant (95% peptide thresholds, 99% protein thresholds) in the sFasL-Fc incubated sample compared with the Fc protein incubated sample. Among them, 29 of unique extracellular space or plasma membrane proteins were selected. Although other candidate proteins could potentially be binding partners of sFasL, a TNF-receptor superfamily protein, we selected TNFRSF10B (DR5) as the top candidate of sFasL binding partners. As shown in Figure 1E, peptide sequence information (LLVPANEGDPTETLR and DASVHTLLDALETLGER) of TNFRSF10B (DR5) were accurately assigned. As pointed out, we added a detailed experimental condition in the Materials and methods section.

2. All commercially purchased reagents that have been used in key experiments, e.g. sFasL, FasL-Fc, TRAIL, anti-Fas, and anti-DR5 antibodies must be characterized and validated in order to confirm that their identity and activity are indeed what they are advertised to be.

To validate these reagents, Jurkat T cells were treated with sFasL, FasL-Fc or TRAIL in the presence or absence of anti-Fas or anti-DR5 antibodies and cell death was estimated. sFasL and TRAIL increased cell death of Jurkat T cells, which was inhibited by adding anti-Fas or anti-DR5 antibodies (Figure 1—figure supplement 3B and C). Moreover, FasL-Fc increased production of CX3CL1 by human synovial fibroblasts as much as sFasL (Figure 3—figure supplement 1B and 1C). These findings indicate that these reagents are validated in terms of functional aspects. These results are presented in Figure 1 —figure supplement 3, and Figure 3—figure supplement 1B and 1C.

3. The specific binding of sFasL to DR5 should be reconfirmed by demonstrating co-immunoprecipitation of the two in endogenous setting, that is to say – using cells that express DR5 at its endogenous levels and using sFaL at concentrations of the recombinant protein that mimic endogenous levels. Assessment of the bioactivity of the ligands should also be done at concentrations of the recombinant protein that are at the range of the endogenous levels.

To address this issue, hFLSCs were pre-treated with recombinant FasL or TRAIL (200 ng/mL, purchased from R&D, 6×His-tagged at N terminus) for 30 min, and incubated with bis (sulfosuccinimidyl) suberate (BS^3^) for cross linking. After then, samples were prepared for immunoprecipitation using ProteoPrep membrane extraction kit (Σ) and immunoprecipitated with anti-DR5 antibody or control IgG, and blotted with anti-FasL, and TRAIL antibody (Figure 1J). On the other hand, to precipitate 6×His-tagged FasL and TRAIL, the samples were also immunoprecipitated with anti-His antibody or control IgG, and blotted with anti-Fas or anti-DR5 antibody Figure 1 —figure supplement 4H. These experiments demonstrated that DR5 bound to both FasL and TRAIL in hFLSCs, which are presented in Figure 1J and Figure 1 —figure supplement 4H.

4. Knockdown does not fully abolish the expression of a protein, nor can the use of antibodies always allow to fully block its activity. To alleviate the concern that such incomplete effects have led to mistaken conclusions the authors are requested to apply CRISPR/Cas9 to knockout the DR5 and Fas genes instead (which is rather easy to do nowadays). They should then further reconfirm the findings by re-expressing the cDNA for the knocked-out genes. Does the knockout of DR5 eliminate a residual response of Fas deficient cells to FasL? Does re-expression of DR5 in such cells rescue such deficiency?

To address this issue, DR5 (TNFRSF10B) and/or Fas (FAS) genes were knockout in human synovial fibroblasts using CRISPR/Cas9 system (Figure 1 —figure supplement 4C). Upon knockout of DR5 or Fas gene in human synovial fibroblasts, sFasL-mediated cell death (Figure 2 —figure supplement 2B) and FasL-Fc binding to cell surface (Figure 1I) were decreased compared with control synovial fibroblasts. When we performed knockout of both Fas and DR5 genes (double knockout; DKO) in human synovial fibroblasts, sFasL-mediated cell death and FasL-Fc binding to cell surface were reduced more in DKO fibroblasts than in fibroblasts knockout either DR5 or Fas gene (Figure 1I). Furthermore, re-expressing DR5 or Fas in DKO synovial fibroblasts restored sFasL-mediated cell death and FasL-Fc binding to cell surface (Figure 1I). However, sFasL-mediated CX3CL1 production was similarly reduced in DR5 gene KO and DKO human synovial fibroblasts, where it is not altered in Fas gene knockout-cells (Figure 3F). Moreover, re-expression of DR5 in DKO cells restored FasL-mediated CX3CL1 production as much as control cells, whereas that of Fas did not alter CX3CL1 production in DKO cells (figure 3F). Based on these findings, it is confirmed that sFasL binds to DR5 and this interaction increases CX3CL1 production by synovial fibroblasts. Moreover, the concern that incomplete effects of blocking antibodies and knockdown system have led to mistaken conclusions has been alleviated.

5. The assessment of the affinity of binding of TRAIL to DR5 should be presented in a figure.

As reviewer suggested, we presented the assessment of the binding affinity of TRAIL to DR5 in Figure 1—figure supplement 4G.

6. Specific triggering of cell death by DR5 in response to sFasL should be validated by the use of inhibitors that specifically affect apoptosis and necroptosis.

As reviewer suggested, we validated DR5-FasL interaction-mediated cell death using specific inhibitors of apoptosis (NSCI for caspase 3) and necroptosis (GSK’872 for RIPK3). Upon treatment of hFLSCs with sFasL, apoptosis and necroptosis were increased, which was inhibited by addition of NSCl and GSK’872, respectively (Figure 2—figure supplement 2A and 2C). In particular, sFasL-induced apoptosis and necroptosis were decreased in Fas or DR5 gene knockout hFLSCs, which was more decreased in DKO (Fas and DR5 gene-knockout) hFLSC (Figure 2—figure supplement 2B and D). These findings indicate that sFasL-DR5 interaction induces both apoptosis and necroptosis in synovial fibroblasts.

(ii) Requests concerning just your way of writing, which must also be met.1. The authors begin the abstract by noting that "Fas-independent membrane bound receptor for FasL has not yet been reported". In the introduction, they argue that "in many studies demonstrating the biological effects of FasL in vitro and in vivo, some of the results have not been confirmed to be Fas-dependent. Thus, it is reasonable to consider the possibility that sFasL may bind Fas-independent receptor in addition to Fas in vivo". The premise that an alternate receptor for sFasL exists is arbitrary: just because this hasn't been ruled out doesn't qualify it as a valid possibility. There has to be some positive evidence in favor of this specific notion in order to qualify it as a starting point or hypothesis for further studies. Please change this statement accordingly.

As reviewer pointed out, we modified and deleted unclear sentences in the abstract and introduction as follows:

Abstract: The interaction of Fas-independent membrane bound receptor and FasL remains elusive.

Introduction: However, whether sFasL-mediated inflammation is regulated in a Fas-independent manner in vivo remains elusive and how it regulates inflammation in various microenvironments is unclear. To address this, we identified a Fas-independent membrane-bound receptor for FasL and explored its functions.

2. Fas lpr mice are not completely negative for Fas. Some tissues actually express wt Fas (PMID: 7528670). That could compound some of the data especially the in vivo data. Please discuss the implications.

As reviewer pointed out, it has been reported that *Fas^lpr/lpr^* mice are not completely negative for Fas expression. Thus, it is reasonable to consider that low levels of Fas might affect sFasL-DR5 interaction-mediated inflammation in joint tissue.

First, Mariani et al. demonstrated that low level of Fas expression was detected in thymocytes from *Fas^lpr/lpr^* mice. However, they have not investigated expression pattern of Fas in synovial fibroblasts. In our experiments, flow cytometric analysis revealed that synovial fibroblasts from *Fas^lpr/lpr^* mice with AIA minimally expressed Fas on cell surface (Figure 1—figure supplement 1A), suggesting that minimal expression of Fas in *Fas^lpr/lpr^* mice might not affect sFasL-DR5 interaction-mediated inflammation in the AIA model.

Second, Fas KO or DR5 KO synovial fibroblasts showed a reduction of sFasL-induced cell death compared with control cells, suggesting that sFasL-Fas interaction-mediated cell death might affect AIA in *Fas^lpr/lpr^* mice. However, our experiments demonstrated that z-VAD-fmk (caspase inhibitor) did not affect AIA in WT mice, although z-VAD-fmk reduced apoptosis of immune cells from the inflamed joints. These findings indicate that sFasL-Fas interaction-mediated apoptosis minimally involved in the pathogenesis of AIA.

Thus, it is unlikely that minimal expression of Fas in *Fas^lpr/lpr^* mice affects AIA. This issue has been discussed in the page 13 as follows: Although Mariani et al. demonstrated that low level of Fas expression was detected in thymocytes from *Fas^lpr/lpr^* mice (Mariani, Matiba, Armandola, and Krammer, 1994), expression pattern of Fas in synovial fibroblasts has been unclear in *Fas^lpr/lpr^* mice. In our experiments, flow cytometric analysis revealed that synovial fibroblasts from *Fas^lpr/lpr^* mice with AIA minimally expressed Fas on cell surface. Moreover, our experiments demonstrated that z-VAD-fmk (caspase inhibitor) did not affect AIA in WT mice, although z-VAD-fmk reduced apoptosis of immune cells from the inflamed joints. Thus, it is unlikely that minimal sFasL-Fas interaction-mediated apoptosis in *Fas^lpr/lpr^* mice affects the pathogenesis of AIA.

3. An interesting consideration is whether the reported increase in metastases in TRAIL-R k.o mice by the Walczak group could be due to FasL unbound to TRAIL-R driving metastasis through increased binding to Fas. In addition, other reports suggested that TRAIL-R deficiency sensitized mice to inflammation, DSS induced colitis or DEN induced liver cancer (PMIDs: 24117005; 18079962). Again, that seems to be at odds with the activity of DR5 described here to promote inflammation. Please discuss the implications.

We would like to appreciate invaluable comment. It is reasonable to consider that sFasL-DR5 interaction might affect various biological events in vivo, including tumor surveillance and inflammation in various organs. Our experiments demonstrated that sFasL-DR5 interaction occurred in the inflamed joint tissue-in particular-synovial fibroblasts, suggesting that this interaction might depend on DR5 and sFasL expression patterns, cell types, and microenvironment status in target tissues. Moreover, the expression of DR5 and FasL has been reported in various cell types, thereby establishing complicated interaction network during various biological events. Thus, it is reasonable that sFasL-DR5 interaction regulates tumor surveillance and inflammation via various ways, which should be investigated further. We discuss this issue in the section of Discussion (page 12 and 13) as follows: In several studies, DR5 KO mice showed enhancement of tumor metastasis and inflammation (Zhu et al., 2014; Finnberg, Klein-Szanto, and El-Deiry, 2008), which appears to be contradictory to inflammatory inducer of sFasL-DR5 interaction. However, the expression of DR5 and FasL has been reported in various cell types (O'Brien et al., 2005; O'Connell, 2000; Yuan et al., 2018), hereby inducing complicated interaction network during biological events, suggesting that sFasL-DR5 interaction regulates tumor surveillance and inflammation via complicated ways, resulting in various biological effects. Thus, these issues should be investigated further.

4. Not once do the authors explain to a wider audience what FLSC stands for.

In the first expression of FLSC, we described FLSC presents fibroblast-like synovial cells in 43 line of page 2 and 68 line of page 3.

5. Figure 1G: How many patients/slides were stained? Statistics need to be provided.

We analyzed DR5-positive cells in the minimum three fields (× 400 magnification) in microsection of joint tissues from three patients. As reviewer suggested, we presented statistics in Figure 1G.

6. Line 104: It should read "Figure 1—figure supplement 2C".

As reviewer pointed out, we modified manuscript.

7. Line 107: It should read "Figure 1—figure supplement 2D and E". 8) Line 133: It should read "Figure 2—figure supplement 1B" instead of Figure S4B.

As reviewer pointed out, we modified manuscript.

8. Figure 3—figure supplement 2 is labeled in the figure legend as Supplemental figure 7, Figure 5—figure supplement 1 as Supplemental figure 9, and Figure 6—figure supplement 1 as Supplemental figure 10, respectively.

As reviewer pointed out, we modified manuscript.

9. The authors state that based on the data sFasL-DR5 interactions induce both apoptosis and necroptosis. Without more detailed analyses it is not clear whether necroptosis is induced. All that the data suggest is that cells undergo secondary necrosis, a phenomenon typical for in vitro stimulated cells. Please discuss.

DR5 activation can be led to two major consequences; inflammation and cell death (apoptosis and/or necroptosis). We tried to figure out which pathway is activated by sFasL-DR5 interactions and exacerbates joint inflammation. To address this issue, specific triggering of cell death by DR5 in response to sFasL was validated using inhibitors that specifically affect apoptosis or necroptosis. Thus, we used specific inhibitors of apoptosis (NSCI for caspase 3) and necroptosis (GSK’872 for RIPK3). Upon treatment of hFLSCs with FasL, apoptosis and necroptosis were increased, which was inhibited by adding NSCI and GSK’872, respectively. In particular, FasL-induced apoptosis and necroptosis were reduced in Fas or DR5 gene- knockout hFLSCs. Moreover, both Fas and DR5 gene-knockout (DKO) hFLSCs showed minimal apoptosis and necroptosis upon sFasL treatment. These findings indicate that FasL-DR5 interaction induces both apoptosis and necroptosis. We discuss this issue in the section of results (page 7) as follows:

Moreover, the sFasL-DR5 and sTRAIL-DR5 interactions similarly induced apoptosis and necroptosis in hFLSCs (von Karstedt, Montinaro, and Walczak, 2017; Wiley et al., 1995), which was inhibited by NSCI (caspase 3 inhibitor) and GSK’872 (RIPK3 inhibitor), respectively (Figure 2F and G, and Figure 2—figure supplement 2A-D). Moreover, FasL-induced apoptosis and necroptosis were partially reduced in Fas or DR5 gene-knockout hFLSCs and almost abolished in both Fas and DR5 gene-knockout (DKO) hFLSCs. These findings indicate that FasL-DR5 and FasL-Fas interactions induce apoptosis and necroptosis.

10. As to the article title. I would suggest to include in the title that this is in the context of autoantibody-induced arthritis. Also the "Fas independent receptor" does not sound good. Something like: Fas ligand drives autoantibody-induced arthritis through binding to DR5/TRAIL-R2

As reviewer suggested, we modified title as follows: Soluble Fas ligand drives autoantibody-induced arthritis via binding to DR5/TRAIL-R2

(iii) Additional requests that I consider to be of secondary importance for the central message of the paper. You might choose either to address these requests or just to delete the data that they concern.1. FasL-Fc is an unnatural construct that (presumably) displays two FasL homotrimers. Rather than using it interchangeably without explanation, the authors should better use just sFasL throughout the study.

We would like to appreciate reviewer’s reasonable comment. First, to address this issue, we biotinylated recombinant sFasL (bio-sFasL) and used this bio-sFasL for flow cytometry experiments. WT and Fas and/or DR5 KO hFLSCs were incubated with bio-sFasL, washed, and treated with streptavidin (sAv)-PE. Flow cytometry demonstrated that WT hFLSCs were stained with sAv-PE, which was decreased in Fas or DR5 KO cells. Moreover, sAv-PE staining was minimally detected in Fas and DR5 DKO cells compared with WT and single gene KO cells (Figure 1—figure supplement 4D). These results were similar to those obtained from experiments using FasL-Fc, suggesting that FasL-Fc is reasonable reagent to estimate binding between FasL and its receptors such as DR5 and Fas.

Second, we validated FasL-Fc in terms of functional aspect by measuring sFasL-mediated cell death and CX3CL1 production of hFLSCs (Figure 3—figure supplement 1B and 1C). FasL-Fc induced cell death and CX3CL1 production of hFLSCs, which were similar to those induced by sFasL (Figure 1 —figure supplement 3D and Figure 3 —figure supplement 1B and C).

Taken together, these findings indicate that biological activities of FasL-Fc are similar to those of sFasL.

2. The authors contrast binding of FasL-Fc to human DR5 with human DR4. What about other TRAIL receptors, and other TNFRSF members? A more complete analysis is called for.

To address this, we knock-downed other TRAIL receptors and TNFRSF members in hFLSCs (Figure 1—figure supplement 4B). However, no membrane receptors were bound to FITC-conjugated FasL-Fc except for Fas and DR5 and produced FasL-mediated CX3CL1 except DR5 (Figure 1 —figure supplement 4A and Figure 3—figure supplement 1I). These findings indicate that FasL binds to Fas and DR5, but not other TNFRSF members.

3. If FasL binds to DR5, then what specific binding site does it occupy? Are there any specific mutations that can disrupt this binding? Is the FasL binding site for DR5 related to TRAIL's? Is there homology between the DR5 and Fas binding sites for FasL?

We presented the results to address these questions in previous version of manuscript (page 7) as follows: Pre-incubation of rhFasL, but not rhTRAIL, reduced rhFasL-Fc protein binding to EL4 cells expressing hFas (Figure 2A and Figure 1—figure supplement 2G), whereas pre-incubation with rhTRAIL or rhFasL inhibited rhFasL-Fc protein binding to hFLSCs and hDR5-expressing EL4 cells, indicating that TRAIL and FasL compete for binding to DR5 (Figure 2A and B, and Figure 1—figure supplement 2F and J). Based on crystal structures of the FasL/DcR3 (PDB 4MSV) and TRAIL/DR5 (1D4V) complexes, FasL forms a trimer similar to other TNF ligands, and DcR3 or DR5 binds to the interface formed by two adjacent FasL or TRAIL monomers, resulting in 3:3 stoichiometry (FasL:DcR3 or TRAIL/DR5) (Figure 2—figure supplement 1B) (Liu et al., 2016; Mongkolsapaya et al., 1999). Moreover, superimposition of these two complexes demonstrates that most of their interaction determinants are similarly arranged. To test whether the binding mode of FasL and TRAIL to DR5 are similar, we mutated amino acids in cysteine-rich domain (CRD)2 or 3 of DR5, which are critical for the TRAIL-DR5 interaction (Figure 2C and Figure 2—figure supplement 1C). In contrast to WT hDR5, rhFasL-Fc protein did not bind to EL4 cells expressing hDR5 mutated in CRD2 or CRD3 (Figure 2D), although the expression levels of WT and mutant hDR5 were similar (Figure 2—figure supplement 1C). Moreover, rhDR5-Fc protein binding was abolished in EL4 cells expressing FasL mutants, which can inhibit the interaction between FasL and DcR3 (Figure 2E and Figure 2—figure supplement 1E). Collectively, these findings indicate that regions of DR5 that bind hFasL may largely overlap with those that bind hTRAIL, and the binding mode of FasL to DcR3 and DR5 is similar, although the precise sites of hFasL and hDR5 interaction remain unclear.

4. All negative data using inhibitors (e.g. zVAD-fmk) need to be supported by positive control data for the inhibitors, particularly in the in vivo experiments.

To address this, we injected mice with zVAD-fmk during AIA and measured cell death of immune cells from joint tissues. Upon injection with zVAD-fmk, the cell death of immune cells was inhibited, indicating that zVAD-fmk acts as caspase inhibitor in vivo. The data are presented in Figure 2 —figure supplement 3B.

5. The authors should provide evidence that sFasL leads to the assembly of a FADD-containing complex that is dependent of DR5 and not Fas using both murine and human Fas and DR5 KO cells.

I would like to appreciate reviewer’s reasonable comment on this issue. However, we did not address this issue further because we decided to delete these data in our manuscript.

6. sFas and TRAIL are known inducers of cell death, however, in the present manuscript the authors do not take into account this variable and the possible effect that cell death might have in, as an example, selecting for a particular cell population. The in vitro experiments should be validated in the presence of cell death inhibitors. In this regard, the authors mention different modes of cell death without testing for them. If such conclusions are to be drawn than the authors should use specific inhibitors of the difference cell death modalities.

To address this issue, specific triggering of cell death by DR5 in response to sFasL was validated by the use of inhibitors that specifically affect apoptosis and necroptosis. Thus, we used specific inhibitors of apoptosis (NSCl for caspase 3) and necroptosis (GSK’872 for RIPK3). Upon treatment of hFLSCs with FasL, apoptosis and necroptosis were increased, which was inhibited by adding NSCI and GSK’872, respectively. In particular, FasL-induced apoptosis and necroptosis were reduced in Fas or DR5 gene- knockout hFLSCs (Figure 2 —figure supplement 2B and 2D). Moreover, both Fas and DR5 gene-knockout (DKO) hFLSC showed minimal apoptosis and necroptosis upon FasL treatment (Figure 2 —figure supplement 2B and 2D). These findings indicate that FasL-DR5 interaction induces both apoptosis and necroptosis but minimally effects on AIA.

7. Throughout the manuscript there are several controls missing. While the authors demonstrate the efficiency of deletion of the siRNAs used, it is imperative to assess the abolishment of the signaling that depends on that receptor in the presence of its respective bona fide ligands. This can be achieved by assessing of the formation of the FADD-containing complex by western blot and downstream output of cell death and gene activation.

To address this issue, we knockout DR5 and/or Fas gene in hFLSC using CRISPR/Cas9 rather than assessing of the formation of the FADD-containing complex using western blot. Flow cytometric analysis demonstrated that the expression of Fas and DR5 on hFLSCs after knockout these genes using CRISPR/Cas9 (Figure 1 —figure supplement 4C). Upon knockout of DR5 or Fas gene in human synovial fibroblasts, FasL-mediated cell death and FasL-Fc binding were decreased compared with WT synovial fibroblasts, which were more decreased by both Fas and DR5 gene knockout (double knockout; DKO) cells. FasL-mediated cell death and FasL-Fc binding were restored in DKO synovial fibroblasts by re-expressing DR5 or Fas. Furthermore, sFasL-mediated CX3CL1 production was reduced in DR5, but not in Fas gene knockout human synovial fibroblasts (Figure 3F). DKO cells also showed a reduction of FasL-mediated cell death and FasL-Fc binding like as DR5 knockout cells. Furthermore, re-expression of DR5 in DKO cells restored FasL-mediated cell death and CX3CL1 production, and FasL-Fc binding, whereas Fas re-expression did FasL-mediated cell death and FasL-Fc binding, but not CX3CL1 production. Thus, both knock-down and knockout system demonstrated similar results in terms of FasL-DR5 interaction and its biological effect on CX3CL1 production by synovial fibroblasts.

8. The western blot presented in figure 3j raises doubt regarding the efficiency of Fas blocking in their experiments. As demonstrated in lanes 7-9 versus lanes 1-3, their blocking antibody for Fas has very little capacity to block sFasL-mediated p-IkBα. The authors should explain this observation and (as specified above) repeat the experiment using KO cells for DR5 and/or Fas.

We would like to appreciate reviewer’s reasonable comment. As reviewer pointed out, we deleted our western data in revised manuscript instead of performing experiments using KO cells for DR5 and/or Fas.

9. Regarding the mechanisms of AIA and its dependency on DR5-sFasL engagement the authors should show the levels of CX3CL1 in mFasL KO mice and in sFasKO mice treated with recombinant sFasL. Additionally, the authors should generate DR5/CX3CL DKO mice and DR5-FasL delt.s DKO to show that these mice do not exhibit AIA upon sFasL treatment.

As reviewer pointed out, we present the levels of CX3CL1 in mFasL KO (*Fasl^Δm/Δm^*) mice and sFasL KO (*Fasl^Δs/Δs^*) treated with recombinant sFasL in the joint tissue during AIA (Figure 4D). Upon induction of AIA, the levels of CX3CL1 were reduced in joint tissues from sFasL KO and tnfrsf10b KO mice, but not mFasL KO mice compared with WT mice. Injection with recombinant sFasL increased the levels of CX3CL1 in the joint tissues from WT, sFasL KO, and mFasL KO mice, but not tnfrsf10b KO mice during AIA.

As reviewer suggested, we have been trying to generate DR5/CX3CL1 and DR5/FasL delt.s DKO mice after editor’s decision that gave us a chance to revise our manuscript. Unfortunately, we have not generated these DKO mice yet due to short of time. Thus, we could not perform experiments using these DKO mice.

10. To show that sFasL-DR5 indeed leads to CX3CL1-induced AIA the authors should add to figure 6 a rescue experiment using sFasL-KO mice and conditionally induce CX3CL1 expression.

In previous version of manuscript, we presented results of rescue experiments using sFasL KO mice. As reviewer pointed out, recombinant CX3CL1 injection restored AIA in sFasL KO mice. Moreover, recombinant sFasL increased the levels of CX3CL1 in the joint tissues of sFasL KO mice (Figure 4D).

[Editors' note: further revisions were suggested prior to acceptance, as described below.]

Comments about the ex vivo data:Specific comments1. The viewers believe that in order to exclude remaining doubts about your claims there is the need for several more rigorous controls, as detailed below:Figure 1HThe figure shows binding of FasL-Fc to DR5-transfected EL4 cells, which is blocked by anti-FasL antibody. There should be a comparison to binding of TRAIL-Fc and to an irrelevant Fc fusion protein. Also, does TRAIL-Fc or anti-DR5 antibody block this FACS shift observed with FasL-Fc? The effect of re-expression of Fas alone, without the re-expression of DR5, in the double knockout cells should also be shown. Similar questions apply to Figure 1I.

I would like to appreciate reviewer’s reasonable suggestion regarding appropriate control for these experiments. However, TRAIL-Fc protein is not commercially available and it is not easy to generate TRAIL-Fc protein by ourselves due to time and technical limitation. Alternatively, we have addressed this point using biotinylated TRAIL, FasL, and TNF-α by biotinylating recombinant proteins. Our experiments demonstrated that biotinylated human FasL (bio-hFasL) and bio-TRAIL bound to hDR5-transfected EL4 cells. Moreover, bio-hFasL binding was blocked by pre-incubation of cells with recombinant human TRAIL or anti-human FasL antibody. However, an irrelevant protein such as biotinylated human TNF-α did not bind to human DR5-transfected EL4 cells. Furthermore, biotinylated human FasL (bio-hFasL) bound to human Fas-transfected EL4 cells, which was also inhibited by pre-incubation of bio-hFasL with recombinant hDR5. In contrast, biotinylated human TRAIL did not bind to human Fas-transfected EL4 cells. These results have been presented in Figure 1H and Figure 1—figure supplement 2I in revised version.

In addition, biotinylated hFasL bound to double knockout cells with re-expression of Fas alone, whereas biotinylated hTRAIL did not, which have been also presented in Figure 1I and Figure 1—figure supplement 4D and E.

Taken together, it has been confirmed that FasL specifically binds to DR5 as well as Fas on the cell surface.

Figure 1I and Figure 1 – —figure supplement 4C and 4DThe authors should provide real-time PCR evidence that the Fas and TRAIL genes have indeed been knocked out in all the cells in the preparations used in these experiments.

As reviewer suggested, we performed real-time PCR for validating knockout of target genes, which has been presented in Figure1—figure supplement 4C.

In Fig, 1JsFasL or sTRAIL is added to hFLSCs, cross-linked, and immunoprecipitated with IgG or anti-DR5 and FasL or TRAIL visualized by immunoblot. PBS is an inadequate control here: they should use another soluble TNFSF ligand as a negative control. Also, the binding of sFasL should be blocked by excess sTRAIL and vice versa.

As reviewer suggested, we performed immunoprecipitation using appropriate control protein such as recombinant TNF-α. Recombinant TNF-α did not bound to immunoprecipitated DR5, whereas recombinant FasL and TRAIL bound to DR5, which have been presented in figure 1J. Furthermore, excess amount (20 times) of recombinant TRAIL inhibited binding of FasL to DR5 and that of recombinant FasL also blocked binding of TRAIL to DR5, which has been presented in Figure 2—figure supplement 1A.

Figure 1KPlease compare binding of DR5-Fc with Fas-Fc. Compare binding to EL4-FasL with binding to EL4-TRAIL (positive) and EL4-mTNFa (negative). Is binding of DR5-Fc blocked by TRAIL or by anti-DR5 antibody?

As reviewer suggested, we compared binding of DR5-Fc and Fas-Fc to human FasL-transfected EL4 cells. Both Fas-Fc and DR5-Fc bound to human FasL-transfected EL4 cells (Figure 1—figure supplement 4J). Furthermore, DR5-Fc bound to human TRAIL or FasL-transfected EL4 cells, whereas it did not bind to human TNF-α-transfected EL4 cells. Compared with DR5-Fc, Fas-Fc bound to human FasL-transfected EL4 cells, but not human TNF-α or TRAIL-transfected EL4 cells. These results have been presented in Figure 1—figure supplement 4K and L. Furthermore, Binding of DR5-Fc to human FasL-transfected EL4 cells were inhibited by pre-incubating with recombinant human TRAIL, anti-human DR5 antibody, or anti-human FasL antibody (Figure 1K).

2. Figure 3H and I and the related supplementary figuresThe relevance of the signalling mechanism by which sCX3CL1 is induced to the rest of the paper is not clear. Moreover, given selectivity caveats with the small molecule the significance of the evidence about this issue is limited. The authors should either delete this part or relate to its implications as just putative.

As reviewer suggested, we described implication of sCX3CL1 signaling mechanism as just putative in abstract, results, and discussion parts as follows:

Abstract; The interaction enhanced *Cx3cl1* transcription and sCX3CL1 generation from FLSCs, which might be in an NF-κB-dependent manner.

Results; These findings indicate that sFasL-DR5 interaction-mediated enhancement of *CX3CL1* transcription and sCX3CL1 generation in hFLSCs, which might be dependent on the NF-κB signaling pathway.

Discussion; In our experiments, the sFasL-DR5 interaction enhanced *Cx3cl1* transcription and sCX3CL1 generation by FLSCs in mice and humans, which might be in an NF-κB dependent manner, whereas the sTRAIL-DR5 interaction did not.

General comments1. How were transfected cell lines generated and validated? Please explain and present the data.

As reviewer pointed out, we have described how to generate transfected cell lines in Materials and methods and validated transfected genes, which have been presented in Figure 1—figure supplement 2H (for *TNFRSF10B* and *FAS* genes) and Figure 1—figure supplement 4 I (for *FASL*, *TNFSF10*, and *TNF* genes).

In page 20, Materials and methods as follows;

Plasmids and transfection

WT human *FAS* (NM_000043.5), *TNFRSF10B* (NM_147187.2), *FASLG* (AY858799.1), *TNF* (NC_000006.12), and *TNFSF10* (NM_001190942.2), and the mutant forms of human *TNFRSF10B* and *FASLG* (as described in Figure 2—figure supplement 1C) were cloned into the pIRES3-puro vector (Clontech). EL4 cells were transfected with these genes using a Neon transfection system kit referred to the conventional protocol (1080 V, 50 ms, 1 time). The expression of transfected genes was examined by real-time PCR (Figure 1 —figure supplement 4I) or flow cytometry (Figure 1 —figure supplement 4K and Figure 2 – —figure supplement 1D and E).

2. The references cited for TRAIL-DR5 interaction and co-crystal structures are incomplete.

As reviewer pointed out, we have added references for TRAIL-DR5 interaction and co-crystal structures as follows;

These findings indicate that TRAIL and FasL compete for binding to DR5. Based on crystal structures of the FasL/DcR3 (PDB 4MSV) and TRAIL/DR5 (1D4V and 1DU3) complexes, FasL forms a trimer similar to other TNF ligands, and DcR3 or DR5 binds to the interface formed by two adjacent FasL or TRAIL monomers, resulting in 3:3 stoichiometry (FasL:DcR3 or TRAIL:DR5) (Figure 2—figure supplement 1A) (Cha et al., 2000; Liu et al., 2016<; Mongkolsapaya et al., 1999).

Comments about the in vivo data:As the authors themselves show (Figure 1—figure supplemental 1A), cells of the lpr mice still express some Fas, although at a greatly reduced level. This, unfortunately, casts some doubt on the interpretation of the in vivo data in this study. The authors should either fully delete all these data or present their implication as just putative. It is a pity that the authors have not used the Fas full knockout mice (which are available from Jackson laboratories). Demonstrating an effect of injected sFasL on those mice would have yielded unequivocal evidence.

As reviewer pointed out, the effect of sFasL-Fas interaction on joint inflammation should be considered in *Fas^lpr/lpr^* mice. Thus, we have presented results and implication of *Fas^lpr/lpr^* mice as putative as follows in results and discussion parts:

Results: Taken together, these findings suggest that sFasL generation in hematopoietic cells promotes AIA by binding Fas-independent receptor, although *Fas^lpr/lpr^* mice have been reported to express low level of Fas in vivo (Mariani, Matiba, Armandola, and Krammer, 1994).

Discussion part: Thus, it is unlikely that minimal sFasL-Fas interaction-mediated apoptosis in Faslpr/lpr mice affects the pathogenesis of AIA, although the effect of sFasL-Fas interaction on arthritis might be completely ruled out in Faslpr/lpr mice.

Comments about the English:The sentence ""the interaction of Fas-independent membrane bound receptor and FasL remains elusive or unclear" makes no sense. It implies the existence of previous evidence of such an alternative receptor. There is no such evidence and hence the starting point of the study is still poorly justified. One could change the sentence in the introduction to "whether sFasL-mediated inflammation is regulated via interaction with Fas in vivo has not been demonstrated directly". Also, the next sentence would make more sense if it started with: "By addressing these questions we identified….".In the abstract the authors claim the following: "Affinity purification mass spectrometry analysis using human fibroblast-like synovial cells (hFLSCs) revealed DR5 as a FasL receptor." This statement is somewhat misleading. As the authors explain in their reply to the reviewers' comments, the mass-spectrometric analysis of the affinity precipitate they obtained with recombinant FasL-Fc, resulted in the identification of many different proteins as putative interactors with DR5 being one of many. It is therefore not correct to claim that the interaction with DR5 would be specific. In the response to the reviewers' comments the authors state that they "selected DR5" from a list of 29 proteins they found in the IP which contained a transmembrane domain or were expressed as extracellular proteins. Whereas this puts the starting point of the study into question (why did they not pick CD70, another TNFRSF member which they found in the list?), I understand the argument but in view of the reviewers they should stay closer to the data and relate to their findings differently. Something along the lines of the following would probably be more appropriate: "Affinity purification mass spectrometry analysis using human fibroblast-like synovial cells (hFLSCs) identified DR5 as one of several candidates as the elusive Fas-independent FasL receptor. Subsequent cellular and biochemical analysis revealed that DR5 can specifically interact with recombinant FasL-Fc protein, albeit with an approximately 60-fold lower affinity than Fas."

I would like to appreciate reviewer’s reasonable comments for several sentences in abstract. As reviewer suggested, we modified several sentences in the abstract as follows:

Whether sFasL-mediated inflammation is regulated via interaction with Fas in vivo has not been demonstrated directly. By addressing these questions we identified FasL specifically interacts with TNFRSF10B, known as DR5. FasL (*Fasl^gld/gld^*)-and soluble FasL (*Fasl^Δs/Δs^*)-deficient mice, but not Fas (*Fas^lpr/lpr^*)-and membrane FasL (*Fasl^Δm/Δm^*)-deficient mice, attenuated autoantibody-induced arthritis (AIA), suggesting sFasL promotes inflammation by binding a Fas-independent receptor. Affinity purification mass spectrometry analysis using human fibroblast-like synovial cells (hFLSCs) identified DR5 as one of several candidates as the elusive Fas-independent FasL receptor. Subsequent cellular and biochemical analysis revealed that DR5 can specifically interact with recombinant FasL-Fc protein, albeit with an approximately 60-fold lower affinity than interaction between TRAIL and DR5.

[Editors' note: further revisions were suggested prior to acceptance, as described below.]

Although you have addressed most of the concerns of the reviewers, the reviewers remained concerned about the lack of a definitive control from the beginning of this review process. This concern has not been addressed yet thoroughly enough. The reviewers maintain their request to provide such control by the use of a full Fas knockout mouse (which are available from Jackson Labs). Therefore, we ask you to obtain these mice and perform the definitive experiment which is to test whether injection of recombinant sFasL in full Fas knockout mice promotes AIA or not.Given that this study may have far-reaching consequences, we believe that the revision with the above experimental results is essential for us to make the final decision.

As Dr. Taniguchi suggested, we obtained frozen embryo of Fas KO mice from RIKEN, Japan and performed experiments after recovering them, which has been taking long time. I would like to deeply appreciate Dr. Taniguchi and reviewers’ kind consideration on work.

In K/BxN serum transfer arthritis model, Fas KO mice showed significant joint inflammation and increased expression levels of *Cx3cl1* as much as wild-type (WT) mice , whereas minimal joint inflammation was found in DR5 KO mice (Figure 1A, and Figure 1—figure supplement 1A, B, and C, and Figure 3C). Furthermore, injection of recombinant sFasL (soluble Fas ligand) enhanced arthritis in WT and Fas KO mice during K/BxN serum transfer arthritis model, whereas it minimally altered joint inflammation in DR5 KO mice (Figure 1—figure supplement 1J). Also, in vitro culture showed that sFasL promoted *Cx3cl1* transcript expression and CX3CL1 secretion by synovial fibroblasts from Fas KO mice (Figure 3G and Figure 3 —figure supplement 1K). Taken together these results and large amount data in our manuscript, we have concluded that sFasL promotes joint inflammation by interaction with DR5, but not Fas.

One additional point: There seems to be a mistake in Figure 1H. The EL4 thymoma cells used in the experiment that was presented in this figure have been shown in quite a number of other studies to express TNF receptors and to respond to TNF. We wonder if the fact that you did not see any sign for binding of biotinylated TNF to the EL4-hDR5 cells that were used in this experiment was due to some fault in the preparation of the biotinylated TNF.

We would like to thank reviewer for reasonable comment. As reviewer suggested, we present new data in Figure 1H.